# Simulations of $^{7}$Be and $^{10}$Be with the GEOS-Chem global model v14.0.2 using state-of-the-art production rates

Minjie Zheng[1,2,3*], Hongyu Liu[4,5], Florian Adolphi[6,7], Raimund Muscheler[2], Zhengyao Lu[8], Mousong Wu[9], and Nønne L. Prisle[3*]

[1]Institute for Atmospheric and Climate Science, ETH Zürich, Zürich, Switzerland
[2]Department of Geology, Lund University, Lund, Sweden
[3]Center for Atmospheric Research, University of Oulu, Oulu, Finland
[4]National Institute of Aerospace, Hampton, Virginia, USA
[5]Science Directorate, NASA Langley Research Center, Hampton, Virginia, USA
[6]Alfred Wegener Institute, Helmholtz Centre for Polar and Marine Research, Bremerhaven, Germany
[7]Faculty of Geosciences, Bremen University, Bremen, Germany
[8]Department of Physical Geography and Ecosystem Science, Lund University, Lund, Sweden
[9]International Institute for Earth System Science, Nanjing University, Nanjing, China

*Correspondence to*: Minjie Zheng (minjie.zheng@env.ethz.ch) and Nønne L. Prisle (nonne.prisle@oulu.fi)

## Abstract

The cosmogenic radionuclides $^{7}$Be and $^{10}$Be are useful tracers for atmospheric transport studies. Combining $^{7}$Be and $^{10}$Be measurements with an atmospheric transport model can not only improve our understanding of the radionuclide transport and deposition processes but also provide an evaluation of the transport process in the model. To simulate these aerosol tracers, it is critical to evaluate the influence of radionuclide production uncertainties on simulations. Here we use the GEOS-Chem chemical transport model driven by the MERRA-2 reanalysis to simulate $^{7}$Be and $^{10}$Be with the state-of-the-art production rate from the CRAC:Be (Cosmic Ray Atmospheric Cascade: Beryllium) model considering realistic spatial geomagnetic cut-off rigidities (denoted as P16spa). We also perform two sensitivity simulations: one with the default production rate in GEOS-Chem based on an empirical approach (denoted as LP67), and the other with the production rate from the CRAC:Be but considering only geomagnetic cut-off rigidities for a geocentric axial dipole (denoted as P16). The model results are comprehensively evaluated with a large number of measurements including surface air concentrations and deposition fluxes. The simulation with the P16spa production can reproduce the absolute values and temporal variability of $^{7}$Be and $^{10}$Be surface concentrations and deposition fluxes on annual and sub-annual scales, as well as the vertical profiles of air concentrations. The simulation with the LP67 production tends to overestimate the absolute values of $^{7}$Be and $^{10}$Be concentrations. The P16 simulation suggests less than 10% differences compared to P16spa but a significant positive bias (~18%) in the $^{7}$Be deposition fluxes over East Asia. We find that the deposition fluxes are more sensitive to the production in the troposphere and downward transport from the stratosphere. Independent of the production models, surface air concentrations and deposition fluxes from all simulations show similar seasonal variations, suggesting a dominant meteorological influence. The model can also reasonably simulate the stratosphere-troposphere exchange process of $^{7}$Be and $^{10}$Be by producing stratospheric contribution and $^{10}$Be/$^{7}$Be ratio values that agree with measurements. Finally, we illustrate the importance of including the time-varying solar modulations in the production calculation, which significantly improve the agreement between model results and measurements, especially at mid- and high- latitudes. Reduced

uncertainties in the production rates, as demonstrated in this study, improve the utility of $^7$Be and $^{10}$Be as aerosol
tracers for evaluating and testing transport and scavenging processes in global models. For future GEOS-Chem
simulations of $^7$Be and $^{10}$Be, we recommend using the P16spa (versus default LP67) production rate.

## 1 Introduction

The naturally occurring cosmogenic radionuclide $^7$Be (half-life of 53.2 days) is monitored worldwide and has
been recognized as a useful tracer in atmospheric dynamic studies (Aldahan et al., 2001; Hernández-Ceballos et
al., 2016; Terzi et al., 2019; Liu et al., 2016). Especially, ratios of radionuclides concentrations with very different
half-lives, such as the $^{10}$Be/$^7$Be ratio, have become powerful tools (e.g., Liu et al., 2022b; Raisbeck et al., 1981)
to disentangle the influence of transport and deposition since both $^7$Be and $^{10}$Be in the troposphere are mainly
removed by wet deposition. In this paper, we aim to improve the utility of $^7$Be and $^{10}$Be as tracers for atmospheric
transport by using state-of-the-art production rates in a global 3-D chemical transport model.
$^7$Be and $^{10}$Be are produced through interactions between atmospheric atoms (mostly oxygen and nitrogen)
and incoming cosmic rays in the atmosphere (Lal and Peters, 1967, referred to as LP67 hereafter; Poluianov et
al., 2016, referred to as P16 hereafter). Due to the atmospheric depth-profile of fluxes of primary cosmic rays, the
formed secondary particles, and their energy, $^7$Be and $^{10}$Be production rates reach their maxima in the lower
stratosphere (Poluianov et al., 2016). About two-thirds of $^7$Be and $^{10}$Be are produced in the stratosphere while the
rest is produced in the troposphere (Poluianov et al., 2016; Heikkilä and Smith, 2013; Golubenko et al., 2022).
Once produced, $^7$Be and $^{10}$Be rapidly attach to aerosol particles and get transported and deposited with their carrier
aerosols by wet and dry depositions (Delaygue et al., 2015; Heikkilä et al., 2013). $^{10}$Be has a half-life of 1.39
million years (Chmeleff et al., 2010) and its decay is thus negligible compared to its average atmospheric residence
time (about 1-2 years) (Heikkilä et al., 2008b). During transport away from the regions of their production, the
$^{10}$Be/$^7$Be ratio increases because $^7$Be decays. The ratio $^{10}$Be/$^7$Be therefore could indicate the path-integrated age
of the air mass. Due to different aerosol residence times in the stratosphere (more than 1 year) and troposphere
(~weeks), the $^{10}$Be/$^7$Be ratio is higher in the stratosphere than in the troposphere. Hence the $^{10}$Be/$^7$Be ratio can be
used to detect the stratosphere-troposphere exchange.
Many studies have focused on understanding the signals in surface $^7$Be measurements from worldwide
monitoring stations (e.g., Hernandez-Ceballos et al., 2015; Rodriguez-Perulero et al., 2019; Uhlar et al., 2020;
Ajtić et al., 2021; Burakowska et al., 2021). Due to the cosmogenic origin of $^7$Be, surface air $^7$Be concentrations
are found to be connected to the 11-year cycle of solar modulation (Leppänen et al., 2010; Zheng et al., 2021a).
In addition, $^7$Be concentrations in the surface air are affected by different meteorological processes depending on
locations, such as stratospheric intrusions (Jordan et al., 2003; Pacini et al., 2015; Yamagata et al., 2019),
scavenging by precipitation (Chae and Kim, 2019; Kusmierczyk-Michulec et al., 2015), vertical transport in the
troposphere (Aldahan et al., 2001; Ajtic et al., 2018; Zheng et al., 2021a) and large-scale atmospheric circulations
(Hernández-Ceballos et al., 2022; Terzi and Kalinowski, 2017).
The ability of general circulation models (GCMs, e.g., GISS ModelE, ECHAM5-HAM and EMAC) and
chemical transport models (CTMs, e.g., GEOS-Chem and GMI ) to capture the main characteristics in $^7$Be and
$^{10}$Be transport and deposition has been demonstrated in previous studies (e.g., Heikkilä et al., 2008b; Koch and
Rind, 1998; Field et al., 2006; Usoskin et al., 2009; Brattich et al., 2021; Spiegl et al., 2022; Liu et al., 2016;
Sukhodolov et al., 2017). For example, Usoskin et al. (2009) found that the influence of the solar proton-induced

<sup>7</sup>Be production peak at the surface in early 2005 is small through the comparison of GISS ModelE simulations and surface air measurements. Heikkilä et al. (2009) showed that stratospheric $^{10}$Be contribution is dominant in the global $^{10}$Be deposition by tracing tropospheric and stratospheric $^{10}$Be separately in the aerosol-climate model ECHAM5-HAM. Spiegl et al. (2022) used the EMAC climate model to investigate the transport and deposition process of $^{10}$Be produced by the extreme solar proton event in 774/5 A.D. They suggested that the downward transport of $^{10}$Be from the stratosphere is mainly controlled by the Brewer-Dobson circulation in the stratosphere and cross-tropopause transport. By comparing the measurements with GEOS-Chem simulations over January-March 2003, Brattich et al. (2021) found that increased $^{7}$Be values in surface air samples in Northern Europe in early 2003 were associated with the instability of the Arctic polar vortex. They also showed that, while the model generally simulates well the month-to-month variation in surface $^{7}$Be concentrations, it tends to underestimate the observations (see their Table 2) partly due to the use of the default LP67 production rate for a solar maximum year (1958) in the GEOS-Chem model (Liu et al., 2001). By using the GMI CTM driven with four different meteorological datasets, Liu et al. (2016) showed that the observational constraints for $^{7}$Be and observed $^{7}$Be total deposition fluxes can be used to provide a first-order assessment of cross-tropopause transport in global models. In comparison to GCMs with or without nudged winds (e.g., Golubenko et al., 2021; Heikkilä et al., 2008b; Spiegl et al., 2022) which involve simulating the entire global circulation and climate, the "offline" CTMs are driven by archived meteorological data sets, either from output of GCMs or from atmospheric data assimilation systems. For example, GEOS-Chem can be driven by the GEOS assimilated meteorology (e.g., MERRA-2 reanalysis data; Gelaro et al., 2017a) or output from the GISS GCM (e.g., Murray et al., 2021).

In comparison with the LP67 production rate using an empirical approach (Lal and Peters, 1967; Liu et al., 2001; Brattich et al., 2021), the recent production models apply full Monte-Carlo simulations of the cosmic-ray-induced atmospheric nucleonic cascade (e.g., Poluianov et al., 2016; Masarik and Beer, 1999). LP67 shows the highest $^{7}$Be and $^{10}$Be production rates compared to other production models (Elsässer, 2013). P16 suggests that LP67 overestimates the $^{7}$Be production rate by 30-50% compared to their production model (Poluianov et al., 2016). Furthermore, the LP67 production rate implemented in GEOS-Chem is only validated for the year 1958, a year with a high solar modulation function (i.e., high solar activity) of 1200 MeV (Herbst et al., 2017). This highlights the problem of quantitatively comparing these uncorrected model outputs with measurements from other time periods. Some studies (e.g., Koch et al., 1996; Liu et al., 2016) have applied a scale factor to account for this solar modulation influence on LP67 production rate. However, this correction is not ideal as the influence of varying solar modulation is latitudinally and vertically dependent. In earlier studies, the $^{10}$Be production rate in GEOS-Chem was simply scaled to the $^{7}$Be production rate based on the ratio estimated from the surface measurements (Koch and Rind, 1998). In addition, $^{10}$Be as simulated by GEOS-Chem has not been evaluated so far. It is hence necessary to update the $^{7}$Be and $^{10}$Be production rates in GEOS-Chem and assess the corresponding impacts on model simulation results.

In this study, we incorporate global $^{7}$Be and $^{10}$Be production rates from the recently published "CRAC:Be" (Cosmic Ray Atmospheric Cascade: Beryllium) model (Poluianov et al., 2016) into the GEOS-Chem model. We simulate $^{7}$Be and $^{10}$Be using GEOS-Chem with the following three production scenarios.

- Scenario I: production rate derived from the "CRAC:Be" model considering realistic geomagnetic cut-off rigidity (P16spa production rate)

• Scenario II: production rate derived from the "CRAC:Be" model considering an approximation of
geomagnetic cut-off rigidities using a geocentric axial dipole (P16 production rate)
• Scenario III: default production rate in GEOS-Chem using an empirical approximation (LP67
production rate)
Scenario I is treated as the standard simulation while the other two are sensitivity tests that also enable
comparison to earlier studies. This paper is organized as follows. Section 2 introduces the GEOS-Chem model
and three different $^7$Be and $^{10}$Be production rates, discusses the methodology and experiment design, and describes
the observational data for model evaluations. In section 3, we first investigate the differences between three
different production scenarios (section 3.1). Then, we evaluate model simulations of $^7$Be and $^{10}$Be with several
published datasets of $^7$Be and $^{10}$Be measurements, in terms of absolute values (section 3.2-3.3), vertical profiles
(section 3.4), and seasonal variations (section 3.6). The budgets and residence times of $^7$Be and $^{10}$Be are given in
section 3.5. We also examine the $^{10}$Be/$^7$Be ratio in the model to assess its ability in capturing the stratosphere-
troposphere exchange (section 3.7). Finally, we investigate the influence of including solar-induced production
rate variability on $^7$Be simulations (section 3.8). Summary and conclusions are given in section 4.
**2 Models and Data**
**2.1 GEOS-Chem model**
GEOS-Chem is a global 3-D chemical transport model (http://www.geos-chem.org) that simulates gases and
aerosols in both the troposphere and stratosphere (Eastham et al., 2014; Bey et al., 2001). It is driven by archived
meteorological data. We use version 14.0.2 (https://wiki.seas.harvard.edu/geos-chem/index.php/GEOS-
Chem_14.0.2) to simulate the transport and deposition of atmospheric $^7$Be and $^{10}$Be. We drive the model with the
Modern-Era Retrospective analysis for Research and Applications, Version 2 (MERRA-2) meteorological
reanalysis (http://gmao.gsfc.nasa.gov/reanalysis/MERRA-2/; Gelaro et al., 2017b). MERRA-2 has a native
resolution of 0.5° latitude by 0.667° longitude, with 72 vertical levels up to 0.01 hPa (80 km). Here the MERRA-
2 data are re-gridded to 4° latitude by 5° longitude for input to GEOS-Chem for computational efficiency.
GEOS-Chem includes a radionuclide simulation option ($^{222}$Rn-$^{210}$Pb-$^7$Be-$^{10}$Be), which simulates transport
(advection, convection, boundary layer mixing), deposition, and decay of the radionuclide tracers (e.g., Liu et al.,
2001; Liu et al., 2004; Zhang et al., 2021a; Yu et al., 2018) . The model uses the TPCORE algorithm of Lin and
Rood (1996) for advection, archived convective mass fluxes to calculate convective transport (Wu et al., 2007),
and the non-local scheme implemented by Lin and Mcelroy (2010) for boundary-layer mixing. As mentioned in
the introduction section, the standard GEOS-Chem model uses the LP67 $^7$Be and $^{10}$Be production rates. After
production, $^7$Be and $^{10}$Be attach to ambient submicron aerosols ubiquitously and their behavior becomes that of
aerosols until they are removed by wet deposition (precipitation scavenging) and dry deposition processes. Note
that neither is the process of attachment explicitly represented nor is the aerosol size distribution considered in the
model. In addition, the decay process is included for the short-lived $^7$Be with a half-life time of 53.2-day. The
decay is minor for the long-living $^{10}$Be, which has a half-life time of 1.39 million years (e.g., Chmeleff et al.,
2010).

Wet deposition includes rainout (in-cloud scavenging) due to stratiform and anvil precipitation (Liu et al.,
2001), scavenging in convective updrafts (Mari et al., 2000), and washout (below-cloud scavenging) by

precipitation (Wang et al., 2011). Scavenged aerosols from vertical layers above are allowed to be released to the atmosphere during re-evaporation of precipitation below cloud. In case of partial re-evaporation, we assume that half of the corresponding fraction of the scavenged aerosol mass is released at that level because some of the re-evaporation of precipitation are due to partial shrinking of the raindrops, which does not release aerosol (Liu et al., 2001) . MERRA-2 fields of precipitation formation and evaporation are used directly by the model wet deposition scheme. Dry deposition is based on the resistance-in-series scheme of Wesely (1989). The process of sedimentation is not included in the model.

To quantify the stratospheric contribution to $^7$Be and $^{10}$Be in the troposphere, we separately transport $^7$Be and $^{10}$Be produced in the model layers above the MERRA-2 thermal tropopause (i.e., stratospheric $^7$Be and $^{10}$Be tracers). This approach was previously used to study cross-tropopause transport of $^7$Be in GEOS-Chem (Liu et al., 2001; Brattich et al., 2021) and Global Modeling Initiative chemical transport models (Liu et al., 2016; Brattich et al., 2017). The stratospheric fractions of $^7$Be and $^{10}$Be are defined as the ratio of the stratospheric $^7$Be and $^{10}$Be concentrations to the $^7$Be and $^{10}$Be concentrations.

## 2.2 $^7$Be and $^{10}$Be production models

The GEOS-Chem currently uses the LP67 production rates of $^7$Be and $^{10}$Be (Lal and Peters, 1967). These production rates are calculated using an analytically estimated rate of nuclear disintegration (stars) in the atmosphere (stars/g air/s), multiplied by the mean production yield of 0.045 atoms/star for $^7$Be and 0.025 atoms/star for $^{10}$Be (Lal and Peters, 1967). These rates are represented as a function of latitude and altitude for the year 1958 and are not time varying.

Here we update the atmospheric $^7$Be and $^{10}$Be production rates in GEOS-Chem with the latest production model: CRAC:Be model by P16 (Poluianov et al., 2016) using the solar modulation function record by Herbst et al. (2017). The solar modulation function record is based on the local interstellar spectrum by Herbst et al. (2017), which was also used in the production model. Given spatially and temporally resolved geomagnetic cut-off rigidities, the P16 model allows the calculation of 3-dimensional, temporally variable $^7$Be and $^{10}$Be production rates, which are necessary for input to atmospheric transport models. The P16 production model is regarded as the latest and one of the most accurate production models for $^7$Be and $^{10}$Be and was used in recent general circulation model simulations (e.g., Golubenko et al., 2021; Sukhodolov et al., 2017).

The production rates of $^7$Be and $^{10}$Be are calculated by an integral of the yield functions of $^7$Be and $^{10}$Be ($Y_i$, atoms g$^{-1}$ cm$^2$ sr), and the energy spectrum of cosmic rays ($J_i$, (sr sec cm$^2$)$^{-1}$) above the cutoff energy $E_c$:

$$Q(\Phi, h, P_c) = \sum_i \int_{E_c}^{\infty} Y_i(E,h) J_i(E, \Phi) \, dE$$

The $i$ refers to different types of primary cosmic ray particles (e.g., proton, alpha and heavier particles). For modelling the contribution of alpha and heavier particles to the total production, their nucleonic ratio in the local interstellar spectrum was set to 0.353 (Koldobskiy et al., 2019). The yield function $Y_i$ is a function of height (h) and kinetic energy per incoming primary nucleon (E) and is directly taken from P16. The energy spectrum of cosmic rays $J_i$ is a function of the kinetic energy (E) and depends on the solar modulation function ($\Phi$)(Herbst et al., 2017). $E_c$ is calculated as a function of the local geomagnetic rigidity cutoff ($P_c$):

$$E_c = E_r\left(\sqrt{1 + \left(\frac{Z_i P_c}{A_i E_r}\right)^2} - 1\right)$$


where $Z_i$ and $A_i$ are the charge and mass numbers of particles, respectively. $E_r$ is the rest mass of a proton (0.938
GeV).

The geomagnetic rigidity cutoff $P_c$ is a quantitative estimation of the Earth's geomagnetic field shielding
effect (Smart and Shea, 2005). Cosmic ray particles with rigidity (momentum per unit charge of the particle)
higher than the geomagnetic cutoff rigidity value can enter the Earth's atmosphere. In several model simulations
of [7]Be and [10]Be (e.g., Field et al., 2006; Koch et al., 1996; Liu et al., 2001), the production is calculated with a $P_c$
simplified as a function of the geomagnetic latitude and geomagnetic dipole moment, called the vertical Stoermer
cut-off rigidity equation (see equation 5.8.2-2 in Beer et al., 2012). However, this is different from the real
geomagnetic cut-off rigidity inferred from the trajectories of particles with different energies using real
geomagnetic field measurements (e.g., Copeland, 2018) which also includes non-dipole moments of the field
(Beer et al., 2012) (Fig. S1). Earlier studies suggested that using the simple centered dipole models (e.g., Stoermer
cut-off rigidity) for cut-off rigidity approximation is limited as they can significantly distort the cut-off rigidity
for some regions (e.g., low-latitude regions) (Pilchowski et al., 2010; Nevalainen et al., 2013)

Here we take the geomagnetic cutoff rigidity from Copeland (2018) that provides the cut-off rigidity at a
fine interval (one degree) in both latitude and longitude. This production rate is denoted as P16spa. To investigate
the effect of this more realistic representation of cut-off rigidity on [7]Be and [10]Be simulations, we also perform
simulations where the cut-off rigidities are approximated by the Stoermer equation (denoted as P16). The
influence of the geomagnetic field intensity variations can be considered negligible on annual and decadal
timescales and are ignored here (e.g. Muscheler et al., 2007; Zheng et al., 2020). It should be mentioned that the
LP67 production is based on an ideal axial dipole cut-off rigidity similar to the P16 production model.

**2.3 GEOS-Chem model experiments and evaluations**
An overview of the performed simulations is shown in Table S1. The simulation with the P16spa production rate
is considered as the standard simulation while the simulations with the P16 and LP67 production rates are
sensitivity tests. The simulation with the P16 production rate is conducted to evaluate the influence of a simplified
approximation of cutoff rigidities resulting from a geocentric dipole. In earlier studies, the LP67 production rate
was used for global model simulations of [7]Be (e.g., Liu et al., 2016; Brattich et al., 2017; Liu et al., 2001; Koch
et al., 1996). The purpose of performing the simulation with the LP67 production rate is to evaluate to what extent
model simulations are biased when applying the default LP67 production. Since the LP67 production rate applies
only for the year 1958 (with a solar modulation function of about 1200 MeV) and does not consider the influences
of the solar variations (e.g., 11-year solar cycle), it underestimates the production rate for the period of 2008-2018
that has an average solar modulation function of 500 MeV. To correct for this solar modulation influence, we
follow the previous studies (e.g., Liu et al., 2016; Koch et al., 1996) by multiplying the model results by a scale
factor of 1.39. It should be noted that this correction is not ideal as the effects of a varying solar modulation on
cosmogenic radionuclide production rates depend on altitude and latitude. All simulations are performed from
2002 to 2018 with the first six-year for spin-up to make sure the $^{10}$Be nearly reaches equilibrium in the atmosphere
and the 2008-2018 period (11 years) for analysis. The simulations are conducted using a 4° latitude × 5° longitude
resolution for computational efficiency (e.g., Liu et al., 2016; Liu et al., 2004).
To evaluate the model's ability to reproduce the variabilities in the observations, we use the statistical
parameters: Spearman correlation coefficients and Root Mean Square Error (RMSE) (Chang and Hanna, 2004).
Spearman rank correlation (R) (Myers et al., 2013) is used as it does not make any assumptions about the variables
being normally distributed. It is less sensitive to outliers in the data compared to the commonly used Pearson
correlation. The fraction of modeled concentrations within a factor of 2 of observations (FA2) is calculated, i.e.,
for which $0.5 < X_{model}/X_{observation} < 2$. Usually, if the scatter plot of the model and measurements is within a
factor of 2 of observations, the model is considered to have a reasonably good performance (e.g., Heikkilä et al.,
2008b; Brattich et al., 2021). For model comparison with surface air concentrations, the model value from the
bottom grid box closest to the corresponding measurement site is selected.

**2.4 $^7$Be and $^{10}$Be observational data for model validation**

The annual mean $^7$Be surface air concentration and deposition measurements are taken from a compilation by
Zhang et al. (2021b). The compilation includes a total of 494 annual mean values for surface air $^7$Be concentrations
and 304 for $^7$Be deposition fluxes. For the deposition measurements, most of them include both wet and dry
deposition, while a few are collected only during rainfall events and thus include only wet deposition. It includes
the data from:

- The Environmental Measurements Laboratory (EML, https://www.wipp.energy.gov/namp/emllegacy/index.htm) Surface Air Sampling Program (SASP), which began in the 1980s,

- The ongoing international monitor program Radioactivity Environmental Monitoring (REM) network (e.g., Hernandez-Ceballos et al., 2015; Sangiorgi et al., 2019),

- International Monitoring System (IMS) organized by the Comprehensive Nuclear-Test-Ban Treaty Organization (CTBTO) (e.g., Terzi and Kalinowski, 2017),

- Some additional datasets in publications not included in the above programs.

We only include the data covering more than 1 year to reduce the influence of inherent seasonal variations. We
further include several recently published data for $^7$Be surface air concentrations and deposition fluxes records
that cover more than 1 year (Burakowska et al., 2021; Liu et al., 2022b; Kong et al., 2022).
The dataset used for investigating the seasonality of $^7$Be surface air concentrations are mainly taken from a
multiyear compilation dataset of IMS from Terzi and Kalinowski (2017). The seasonal $^7$Be deposition data are
taken from Courtier et al. (2017), Du et al. (2015), Dueñas et al. (2017), Hu et al. (2020), Lee et al. (2015), and
Sangiorgi et al. (2019). The vertical profile of $^7$Be concentrations is taken from the Environmental Measurements
Laboratory (EML) High Altitude Sampling Program (HASP) spanning the years of 1962-1983. It should be noted,
different from surface air measurements, the vertical air samples were usually collected during single-day flight
campaigns.
There are fewer $^{10}$Be measurements compared to $^7$Be. Here we compiled two datasets of published $^{10}$Be
surface air measurements (Table S2) (Aldahan et al., 2008; Liu et al., 2022a; Yamagata et al., 2019; Padilla et al.,

2019; Rodriguez-Perulero et al., 2019; Huang et al., 2010; Méndez-García et al., 2022; Elsässer et al., 2011; Dibb et al., 1994) and deposition fluxes (Table S3) covering more than 1 year, to validate the model performance. The air samples are continuously collected by filters using a high-flow aerosol sampler. The sampling volume is approximately 700 $m^3$ of air for daily samples (e.g., Liu et al., 2022a) and between 3000 $m^3$ and 5000 $m^3$ for weekly samples (e.g., Yamagata et al., 2019). The deposition data include the precipitation samples (wet deposition) (Graham et al., 2003; Monaghan et al., 1986; Somayajulu et al., 1984; Heikkilä et al., 2008a; Raisbeck et al., 1979; Maejima et al., 2005) and ice core samples (wet and dry deposition) that cover the recent period (Heikkilä et al., 2008a; Zheng et al., 2021b; Pedro et al., 2012; Baroni et al., 2011; Aldahan et al., 1998; Berggren et al., 2009; Auer et al., 2009; Zheng et al., 2023b). The $^{10}Be$ vertical profile measurements are mainly taken from Dibb et al. (1994, 1992) and Jordan et al. (2003).

## 3 Results and Discussions

### 3.1 $^7Be$ and $^{10}Be$ production rates

Figure S2 shows the comparison between $^7Be_{P16}$ and $^7Be_{LP67}$ production rates for the year 1958. Generally, the $^7Be_{P16}$ production rate shows a similar production distribution as the $^7Be_{LP67}$ production rate, with a maximum $^7Be$ production over the polar stratosphere (~100 hPa). The $^7Be_{LP67}$ production rate shows, on average, about 72% higher production rate compared to $^7Be_{P16}$ in the stratosphere and about 38% in the troposphere (Fig. S2c; Table S4). On a global average, the $^7Be_{LP67}$ production rate is about 60% higher than that of $^7Be_{P16}$ as shown in previous studies (Poluianov et al., 2016). The stratospheric production contributes about 67% to the total production for the $^7Be_{LP67}$ production rate while it is about 62% for the $^7Be_{P16}$ production rate for the year 1958.

The $^{10}Be_{LP67}$ production rate in the GEOS-Chem model uses the identical source distribution as $^7Be$ with a scaling factor based on the estimates from surface air measurements (Koch and Rind, 1998). This leads to a constant $^{10}Be_{LP67}/^7Be_{LP67}$ production ratio (0.55) throughout the entire atmosphere. However, as shown in many $^7Be$ and $^{10}Be$ production models (e.g., Poluianov et al., 2016; Masarik and Beer, 2009), $^7Be$ and $^{10}Be$ have different altitudinal production distributions. The P16 production shows an increasing $^{10}Be/^7Be$ production ratio from higher altitude (0.35) to lower altitude (0.6) (Fig. S3). Using a constant $^{10}Be/^7Be$ production ratio may thus result in large errors in the modeled $^{10}Be$ concentrations as well as $^{10}Be/^7Be$ ratios. The stratospheric production contributes about 67% of the total production with $^{10}Be_{LP67}$ while it is about 58% with the $^{10}Be_{P16}$ production for the year 1958 (Table S4).

Figure 1 shows the comparison between $^7Be_{P16}$ and $^7Be_{P16spa}$ production rates for the period 2008-2018. The global production is similar for P16spa and P16 (Table S4). However, considering non-dipole moment influence on geomagnetic cut-off rigidity, $^7Be_{P16spa}$ and $^{10}Be_{P16spa}$ production rates in the Southern Hemisphere show ~11% higher production rates compared to the Northern Hemisphere (Table S4). This difference is not present when an axial dipole is assumed. Compared to P16 production rate, the $^7Be_{P16spa}$ production rate shows 30-40% lower production over eastern Asia and southeastern Pacific, but 40-50% higher over North America and from subtropical South Atlantic to Australia (Fig. 1). $^{10}Be_{P16spa}$ shows similar results as the $^7Be_{P16spa}$. These differences are not constant throughout the atmospheric column but generally increase with altitude (Fig. 1d).

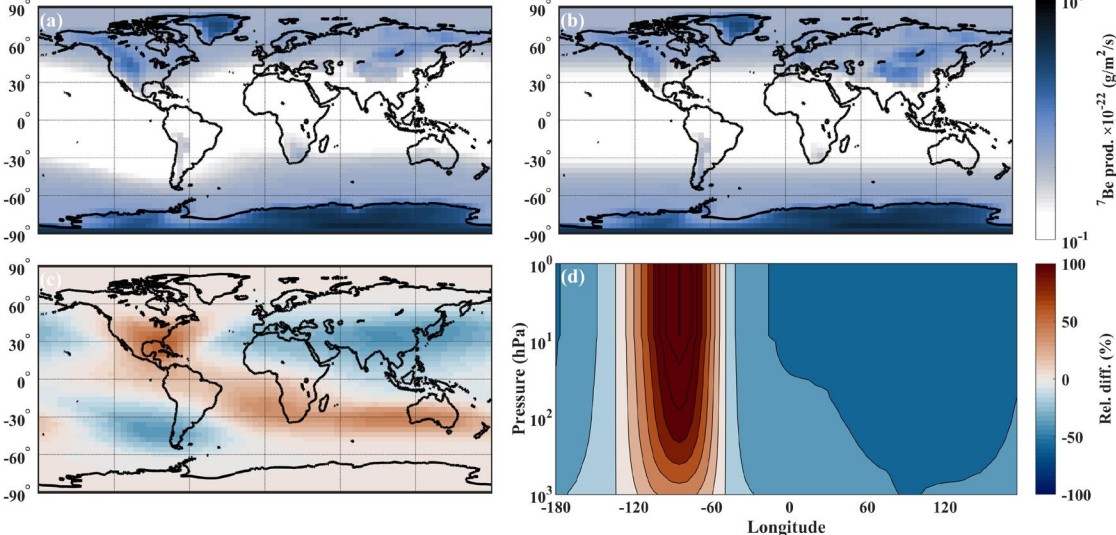

**Figure 1**. Upper panels: Spatial distribution of (a)$^7Be_{P16spa}$ and (b)$^7Be_{P16}$ production rates at 825 hPa over the period 2008-2018. Lower panels: (c) Relative differences (%), i.e., ($^7Be_{P16spa}$-$^7Be_{P16}$)/ $^7Be_{P16}$ ×100%, between production rates with and without considering the detailed spatial cut-off rigidity. (d) Relative differences (%) of the zonal mean production rates between P16spa and P16 at 30°N.

## 3.2 $^7$Be surface air concentrations and deposition fluxes

Figure 2 compares the simulated $^7Be_{P16spa}$ averaged over 2008-2018 with the measurements. Due to the data availability, the measurements do not necessarily cover the same period as model simulations. The model deposition fluxes here include both dry and wet deposition. About 93.7% of modeled air $^7Be_{P16spa}$ concentrations agree within a factor of 2 with the observed values. The model also shows reasonable agreement with the measured deposition fluxes (60.9% within a factor of 2) although the discrepancy between the modeled and observed deposition fluxes is larger than that for surface air concentrations. The deposition fluxes are usually less well monitored compared to the air $^7$Be samples and cover usually only shorter periods (e.g., one or two years). Further, the limited model resolution applied here may not be able to capture meteorological conditions on local scales (e.g., precipitation, convection, and tropopause folding) in some sites (e.g., Yu et al., 2018; Spiegl et al., 2022), especially for coastal regions when the sub-grid scale orographic precipitation is important.

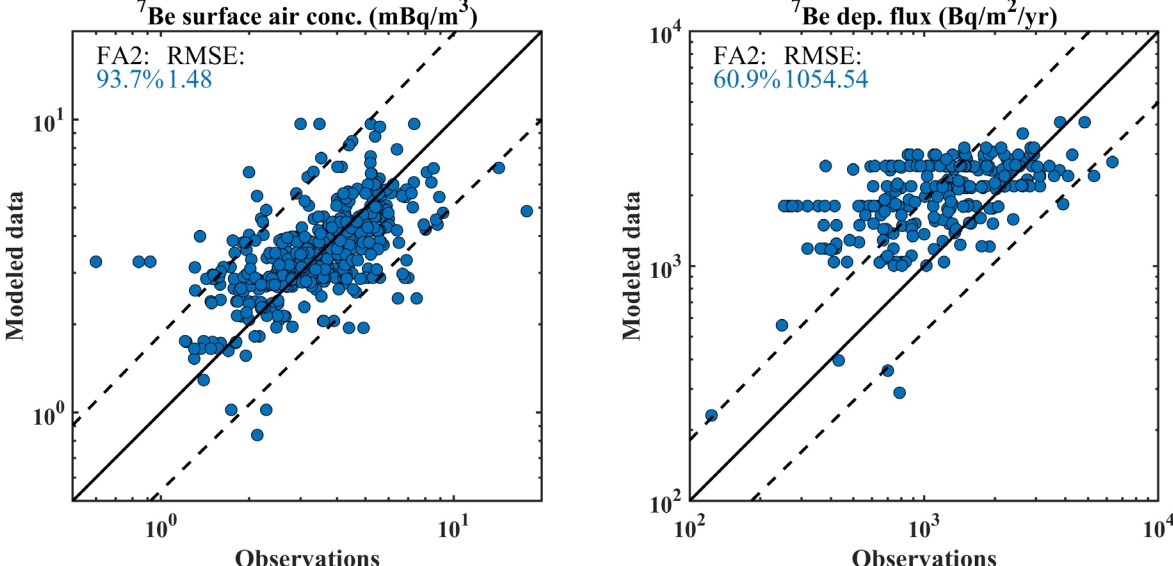

**Figure 2**. Scatter plot of modeled $^7Be_{P16spa}$ versus observed $^7Be$ surface air concentrations (left panel) and deposition fluxes (right panel). The model values are averaged over the years of 2008-2018. The dashed lines are the factor of 2 of 1:1 line (straight lines). The "FA2" indicates the fraction of modeled concentrations within a factor of 2 of observations while "RMSE" indicates the root mean square error.

Figure 3 shows the spatial distribution and zonal mean of measurements in comparison with the model simulated $^7Be_{P16spa}$ surface air concentrations and deposition fluxes. Generally, the model captures the spatial distribution of $^7Be$ air concentrations and deposition fluxes. The "latitudinal pattern" of surface air $^7Be$ concentrations differs from that of $^7Be$ production rate, reflecting the effects of atmospheric transport and deposition processes. The model suggests high $^7Be$ air concentrations mainly over the dry regions (Fig. 3a) due to low wet deposition rates (e.g., desert regions over Northern Africa, Arabian Peninsula, central Australia, and central Antarctica) and over high-altitude regions (e.g., Tibetan Plateau). The model captures the observed latitudinal peaks in surface air concentrations over the subtropics and mid-latitudes (Fig. 3c around 30°N-40°N and 30°S -40°S). These peaks are consistent with the high stratospheric contribution (25%-30%) at mid-latitudes (Fig. S4). The model overestimates $^7Be$ air concentrations over the Arctic (70°N-90°N, Fig. 3c) by about 30%-40%. By contrast, high $^7Be$ deposition fluxes are observed at mid-latitudes due to the influence of the high precipitation (wet deposition) and strong stratosphere-troposphere exchange (Fig. 3d). In the Northern Hemisphere, the model simulated deposition fluxes peak at a lower latitude (~30°N) relative to the observations (~45°N). These modeled spatial distributions of the air concentrations and deposition rates of $^7Be$ also agree generally well with previous model simulations (e.g., Heikkilä and Smith, 2012).

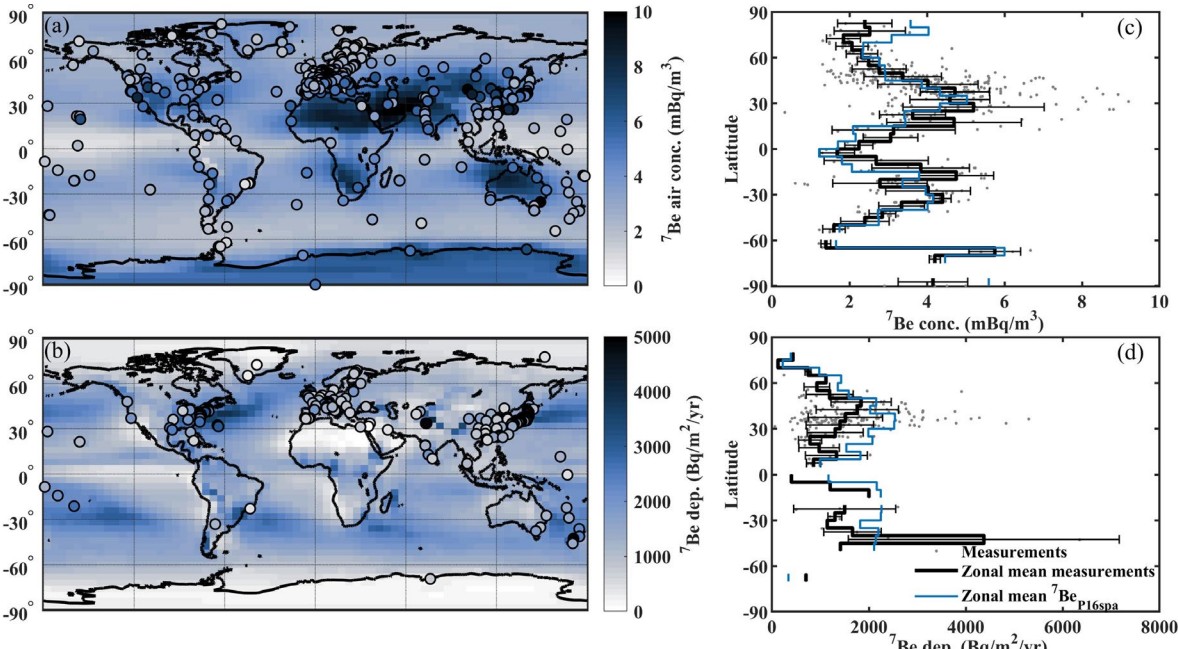

**Figure 3**. Left column: (a) modeled $^7Be_{P16spa}$ surface air concentrations (mBq/m$^3$) and (b) deposition fluxes (Bq/m$^2$/yr) averaged over the period 2008-2018. Color-coded dots denote $^7Be$ measurements. Right column: zonal mean of (c) observed $^7Be$ surface air concentrations and (d) deposition fluxes (black lines, for each 5° latitude bin) compared with the model simulation using the P16spa production rate (blue lines). Dots are individual measurements. The error bars indicate one standard deviation. The outliers, defined as more than three scaled median absolute deviations (MAD) away from the median, are excluded from the calculation. The observations are averaged over the years available.

The modeled $^7Be_{P16spa}$ air concentrations show better agreements (smaller RMSE and higher FA2 values) with the measurements in comparison to $^7Be_{LP67}$ (Fig. S5). $^7Be_{LP67}$ tends to overestimate the absolute values of $^7Be$ concentrations. This is caused by i) the overestimation of $^7Be$ production rate by LP67 for a given solar modulation function and ii) using a simple scale factor to account for the solar modulation influence on the LP67 $^7Be$ production rate.

We also examine whether using the dipole-approximation of the cut-off rigidity or real cut-off rigidity (P16 and P16spa, respectively) in the production model leads to significantly different results (Fig. 4). Although large regional differences (up to 40-50%, Fig. 1) in the production model are observed between P16spa and P16 production rates, such differences are reduced in surface air concentrations and deposition fluxes due to transport and deposition processes, as expected. The $^7Be_{P16sap}$ air concentrations show higher values (~7%) over 10°S-40°S and lower values (~12%) over the east Asian region (Fig. 4) compared to $^7Be_{P16}$. These differences are higher for the deposition fluxes with up to 10% higher over the 10°S-40°S and up to 18% lower over the east Asian region (Fig. 4). Since the total deposition flux reflects precipitation scavenging through the tropospheric column, it tends to be more sensitive to $^7Be$ air concentrations at higher altitudes and downward transport of $^7Be$ from the stratosphere. Indeed, model results suggest that deposition fluxes have a higher stratospheric fraction compared to surface air concentrations (Fig. S4), as previously shown by Liu et al. (2016). The $^7Be_{P16spa}$ deposition fluxes show better agreement with measurements than those of $^7Be_{P16}$ (Fig. S5). The comparison for $^{10}Be$ shows similar results as $^7Be$ except with less than 10% differences. For $^{10}Be$ deposition fluxes in Antarctica and Greenland, this influence is less than 3%. This is because the dominant contribution of $^{10}Be$ is from the stratosphere where the hemispheric production differences are diminished by the long stratospheric residence time of $^{10}Be$. However, it does not suggest that the cut-off rigidity including the non-dipole influence could be ignored for $^{10}Be$ depositions in polar regions, as the spatial pattern of cut-off rigidities was very different in the past time, e.g., during the

Laschamps geomagnetic field minimum around 41,000 years before the present (Gao et al., 2022). Further studies
are warranted to investigate this spatial cut-off rigidity influence on $^{10}$Be in more detail.

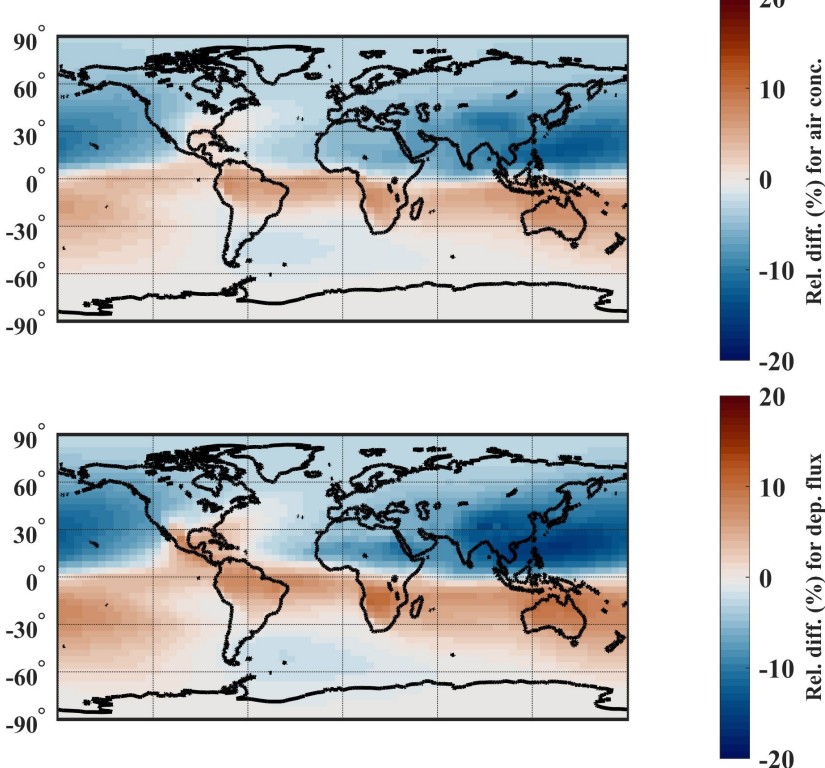

**Figure 4**. Relative differences (percentage) of surface air concentrations (upper panel) and deposition fluxes
(lower panel) between $^{7}$Be$_{P16spa}$ and $^{7}$Be$_{P16}$ for the period 2008-2018, i.e., ($^{7}$Be$_{P16spa}$-$^{7}$Be$_{P16}$)/ $^{7}$Be$_{P16}$ ×100%.

### 380 3.3 $^{10}$Be surface air concentrations and deposition fluxes

Figure 5 shows the comparison between modeled annual mean $^{10}$Be$_{P16spa}$ surface air concentrations (or deposition
fluxes) averaged over 2008-2018 and measurements. The $^{10}$Be$_{P16spa}$ shows similar spatial distributions as $^{7}$Be$_{P16spa}$
because both radionuclides share the same transport and deposition processes. The model underestimates the
measured $^{10}$Be surface air concentrations and deposition fluxes at some sites (Fig. 5b, 5d). This may be attributed
to the influence of resuspended dust with $^{10}$Be attached, which could typically contribute 10%-35% to the air $^{10}$Be
concentrations (Monaghan et al., 1986). It should be mentioned that $^{7}$Be decays in the dust because of its short
half-life, and therefore does not contribute to the surface air $^{7}$Be concentrations. Indeed, data where a careful
examination of the recycled dust $^{10}$Be in samples was conducted (e.g., Monaghan et al., 1986), or from locations
that are less influenced by recycled dust $^{10}$Be (e.g., Polar regions; dots in Fig. 5b-5d), show better agreement with
the model simulations. This suggests the importance of considering the dust contribution when measuring the air
$^{10}$Be samples. The model also shows relatively good agreement with most $^{10}$Be deposition data from polar ice
cores (marked as dots in Fig. 5d) within a factor of 2.

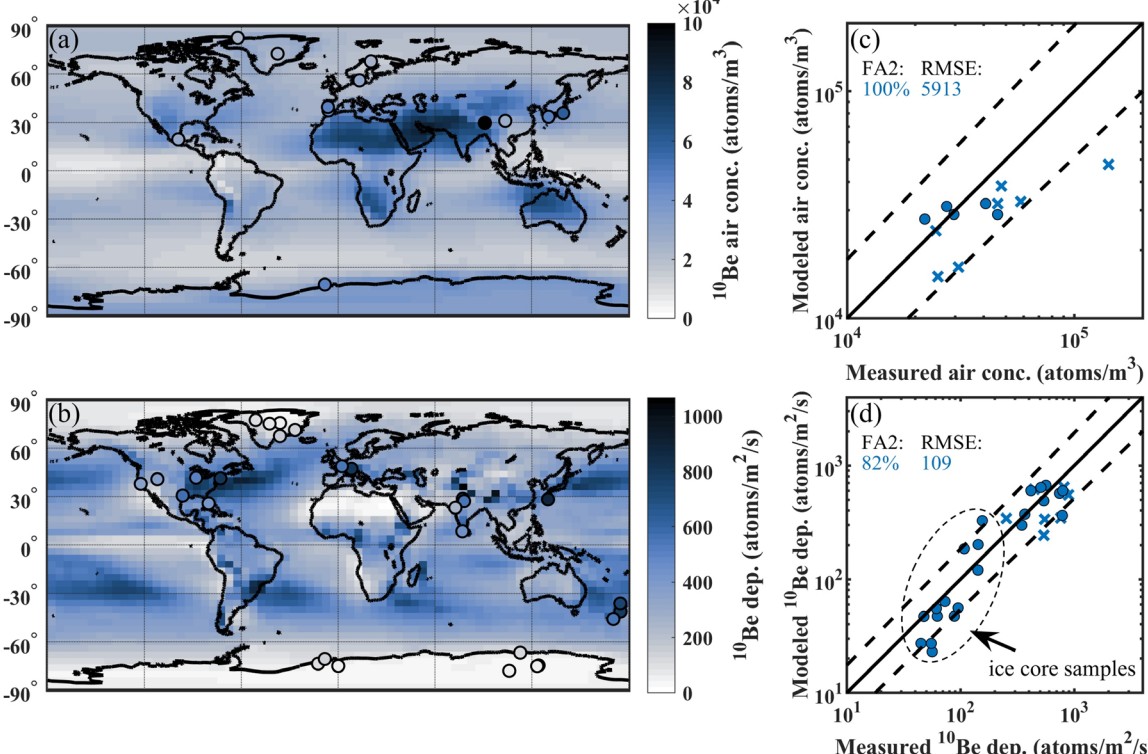


**Figure 5**. Left column: the modeled annual mean $^{10}$Be$_{P16spa}$ (a) surface air concentrations and (b) deposition fluxes averaged
over 2008-2018 overplotted with measurements (color-coded dots). Right column: (c)-(d) the scatter plot between model
results and measurements for (c) surface air concentrations and (d) deposition fluxes. The dots in (c-d) indicate measurements
with careful examination of dust $^{10}$Be contributions or from the polar regions which are not influenced by dust $^{10}$Be. The
crosses indicate the samples without examining dust contributions. The FA2 and RMSE are calculated only using the dust-
free samples (dots). Blue and orange colors indicate the results using P16spa and LP67 production rates, respectively.

**3.4 Vertical profiles of $^7$Be and $^{10}$Be**
Figure 6 shows the simulated annual zonal mean vertical profiles of $^7$Be$_{P16spa}$ and $^{10}$Be$_{P16spa}$ concentrations
compared with those from aircraft measurements in the troposphere and stratosphere from the EML/HASP. The
measurements cover different regions and specific meteorological conditions; hence they should only provide a
range in which the model results should lie. Following previous modelling studies (Heikkilä et al., 2008b; Koch
et al., 1996), we compare model zonal mean values in each 15° latitude band with the corresponding observations.
The simulated $^7$Be$_{P16spa}$ profiles agree well with the measurements, especially capturing the peaks at ~20-22
km at mid- and low- latitudes (e.g., Fig. 6c, 6e, 6h). The feature that $^7$Be increases with altitude without a peak at
22 km at northern high latitudes (60°N-75°N) is also captured by the model (Fig. 6a). The $^7$Be$_{P16spa}$ shows high
concentrations in the polar stratosphere and low values over the equatorial stratosphere (Fig. S6), mainly reflecting
the latitudinal distribution of the production. This "latitudinal structure" is modulated for $^{10}$Be$_{P16spa}$ in the
stratosphere as $^{10}$Be is better mixed than $^7$Be due to its slow decay together with relatively long residence time in
the stratosphere (Waugh and Hall, 2002). Both $^7$Be and $^{10}$Be show very low concentrations in the tropical upper

troposphere, reflecting the frequent injection of air from the lower troposphere in wet convective updrafts, where aerosols are efficiently scavenged (Fig. S6).

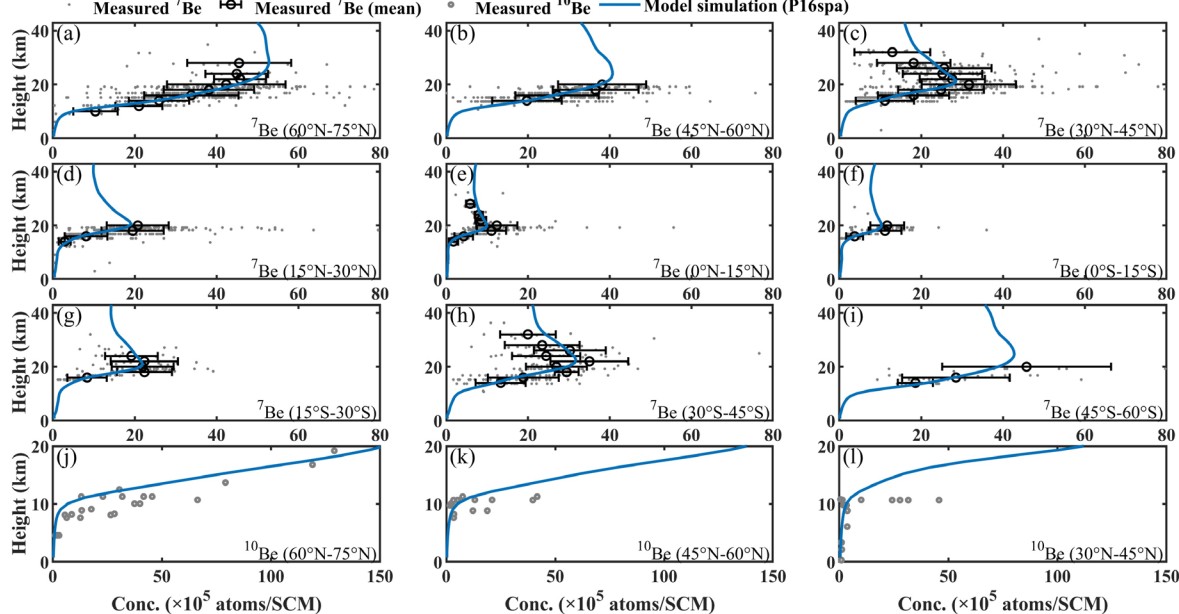

**Figure 6**. Comparison of the vertical profile between measurements (circles) and model zonal mean $^7Be_{p16spa}$ and $^{10}Be_{p16spa}$ concentrations for each latitudinal band (15°) over the period 2008-2018. The $^7Be$ (circle with error bar) observations (from the EML/HASP) are averaged for the altitude band of every 2 km where more than 5 samples are available. We exclude the outlier from the calculation, which is defined as more than three scaled median absolute deviations (MAD) away from the median. The $^{10}Be$ profile measurements are mainly taken from Dibb et al. (1994, 1992) and Jordan et al. (2003).

The model also reasonably simulated $^{10}Be$ vertical profiles compared with observations, with a tendency to underestimate observations in the stratosphere (Fig. 6j-6l). A previous general circulation model study by Heikkilä et al. (2008b) also showed too low model stratospheric $^{10}Be$ compared to measurements. They attributed this underestimation to too short stratospheric air residence time in the model, which prevents $^{10}Be$ concentrations from sufficiently accumulating in the stratosphere. However, this may not be the case in our study, as the stratospheric air residence time in the MERRA-2 reanalysis agrees reasonably with the observations (Chabrillat et al., 2018). Another explanation is that the $^{10}Be$ production rate may be underestimated in the stratosphere. $^7Be$ is less affected by this process than $^{10}Be$ because of its short half-life compared to its stratospheric residence time (Delaygue et al., 2015).

### 3.5 Global budgets and residence time

Table 1 shows the global budgets for $^7Be_{P16spa}$ and $^{10}Be_{P16spa}$ over the period of 2008-2018. About 22.1% of tropospheric $^7Be_{P16spa}$ is lost by radioactive decay, 75.8% by convective and large-scale precipitation, and 2.1% by dry deposition. The wet deposition contributes to about 97% of total deposition for $^7Be_{P16spa}$ and $^{10}Be_{P16spa}$ (Table 1; Fig. S7), which is slightly higher than the ~93% contribution in previous model studies (Heikkilä et al., 2008b; Koch et al., 1996; Spiegl et al., 2022). The global mean tropospheric residence time of $^7Be_{P16spa}$ is about 21 days, which is comparable to those reported by previous model studies: 18 days by Heikkilä et al. (2008b) and

21 days by Koch et al. (1996) and Liu et al. (2001). This also agrees with the residence time of about 22-35 days
estimated from the observed deposition fluxes and air concentrations at 30°N - 75°N (Bleichrodt, 1978). The
averaged tropospheric residence time of $^{10}Be_{P16spa}$ is about 24 days, which is consistent with the 20 days suggested
by Heikkilä et al. (2008b).

**Table 1**. Global budgets of $^{7}Be$ and $^{10}Be$ averaged over the period 2008-2018 in GEOS-Chem using P16spa.

| | $^{7}Be_{P16spa}$ | $^{10}Be_{P16spa}$ |
|---|---|---|
| Sources (g d-1) | 0.403 | 0.256 |
| Stratosphere | 0.272 (67.5%) | 0.161 (62.9%) |
| Troposphere | 0.131 (32.5%) | 0.095 (37.1%) |
| Sinks (g d-1) | 0.404 | 0.253 |
| Dry deposition | 0.004 (1.0%) | 0.006 (2.4%) |
| Wet deposition | 0.151 (37.4%) | 0.247 (97.6%) |
| Radioactive decay | 0.249 (61.6%) | --- |
| Stratosphere | 0.205 (50.7%) | --- |
| Troposphere | 0.044 (10.9%) | --- |
| Burden (g) | 19.145 | 89.902 |
| Stratosphere | 15.778 (82.4%) | 83.785 (93.2%) |
| Troposphere | 3.367 (17.6%) | 6.117 (6.8%) |
| Tropospheric residence time (days)* | 21.72 | 24.08 |

    *Against deposition only


## 3.6 Seasonality in $^{7}Be$ and $^{10}Be$

The seasonality of $^{7}Be$ is influenced by a) the amount of precipitation; b) the stratosphere-troposphere exchange
processes; and c) the vertical transport of $^{7}Be$ in the troposphere. The roles of these factors may vary depending
on location. We compare the seasonal variations of modeled $^{7}Be_{P16spa}$ and $^{7}Be_{LP67}$ concentrations with
measurements from a dataset compiled by Terzi and Kalinowski (2017) with the data covering more than 6 years
(Fig. 7). It should be noted that the model $^{7}Be$ results and MERRA-2 precipitation rates are averaged over the
years of 2008-2018 while the measurements are based on the data availability over the period 2001-2015.

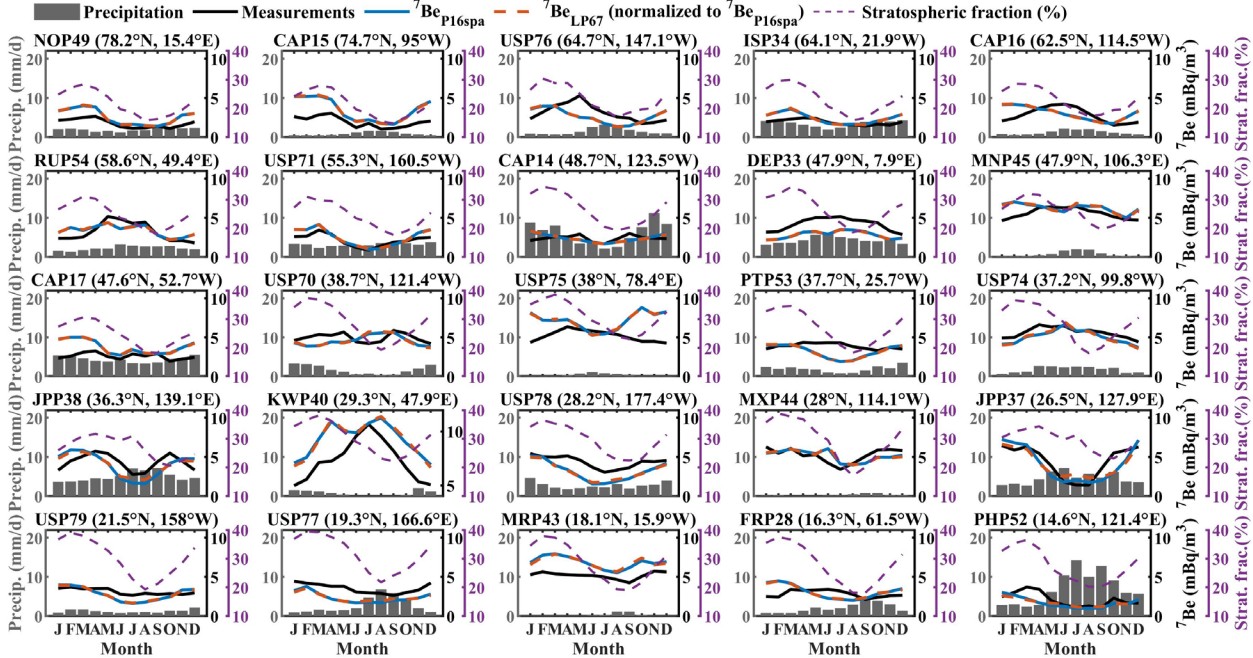

**Figure 7**. Seasonal cycle of simulated and measured surface air $^7$Be concentrations, MERRA-2 total precipitation (4° × 5°, bar graph), and modeled stratospheric contributions to surface air. The plots are arranged based on the site latitudes. The model results using the LP67 production rate are normalized to the ones using the P16spa production rate.

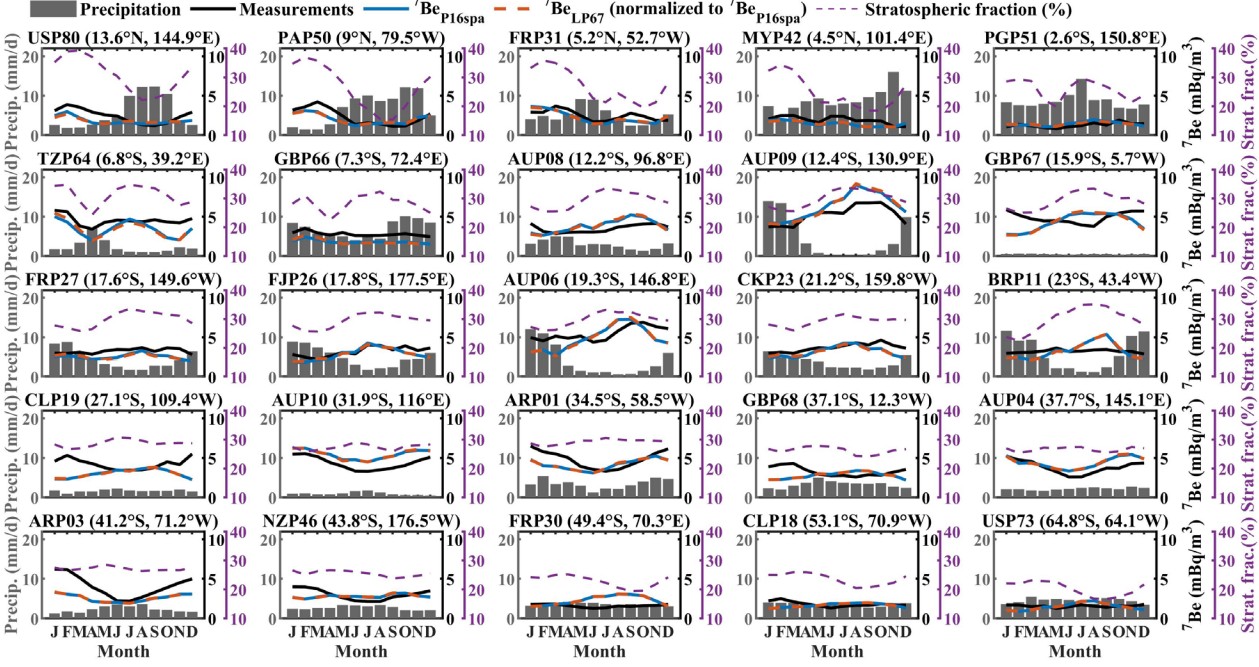

**Figure 7**. (continued)

In the Southern Hemisphere from 25°S-40°S, the $^7$Be concentration peak is observed in austral summer (December-February), resulting from the combined influence of stratospheric intrusions and strong vertical transport during this season (Villarreal et al., 2022; Zheng et al., 2021a; Koch et al., 1996). The summer peak is also observed at northern mid-latitudes. This "summer peak" feature is well simulated by the model at some sites (e.g., KWP40 (29.3°N, 47.9°E), AUP04 (37.7°S, 145.1°E) and AUP10 (31.9°S, 116°E) shown in Fig. 7) but not

at others (e.g., GBP68 (37.1°S, 12.3°W) and PTP53 (37.7°N, 25.7°W) in Fig. 7). This may not be related to
stratospheric intrusion in the model as the simulated stratospheric contributions (Fig. S4) agree fairly well with
estimates inferred from measurements, i.e., ~25% on annual average at northern mid-latitude surface (Dutkiewicz
and Husain, 1985; Liu et al., 2016). Hence this could be due to the errors in vertical transport (e.g., convection)
during the summer season.

The sites at northern high-latitudes (>50°N) show spring peaks that are well simulated by the model (e.g.,

ISP3 (64.1°N, 21.9°W)). This spring peak coincides with high stratospheric contributions, reflecting the influence
of stratospheric intrusions. The influence of precipitation changes is also seen at several sites, especially in
locations with high precipitation rates (e.g., monsoon regions). For example, two sites from Japan (JPP38 (36.3°N,
139.1°E) and JPP37 (26.5°N, 127.9°E) in Fig. 7) show summer minima coinciding with the high precipitation,
even with relatively high stratospheric contributions in the same month.

The seasonal variation of stratospheric contribution is quite similar for the sites located in the Northern

Hemisphere, with a high contribution in spring and a low contribution in fall. This is consistent with the estimates
based on air samples that indicate stratospheric contributions varying from ~40% in spring to ~15% in fall at
latitudes 38°N-51°N (Dutkiewicz and Husain, 1985).

Generally, the model simulates well the annual cycle of surface air $^{7}$Be concentrations for most sites in terms

of amplitude and seasonality (Fig.7). For a few sites (e.g., DEP33 (47.9°N, 7.9°E)), the model captures the
observed seasonality but not the correct absolute values. This could be partly due to the coarse resolution of the
model. The $^{7}$Be$_{LP67}$ is normalized to $^{7}$Be$_{P16spa}$ as we focus on the comparison of seasonal variability between these
simulations. The very similar features (differences within 1%) between all simulations using different production
rates indicate a dominant influence of the meteorological conditions on the seasonal variations of the air $^{7}$Be
concentrations.

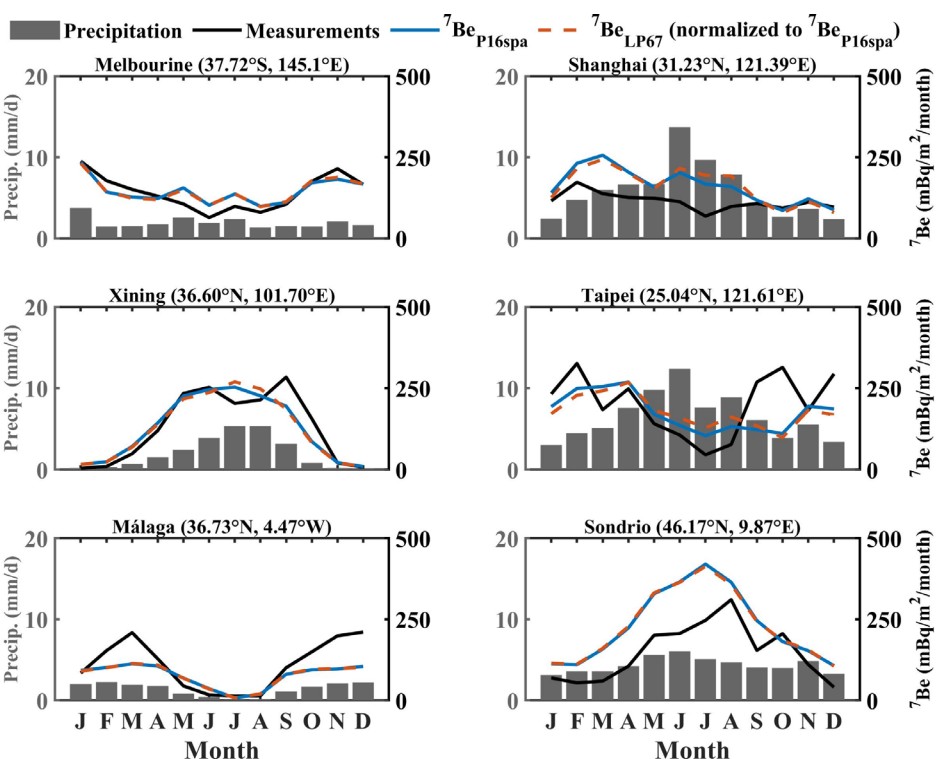


Figure 8 compares model results with the seasonal $^7$Be deposition flux observations over the overlapping periods. Usually, high precipitation leads to high $^7$Be deposition fluxes (e.g., Du et al., 2015). Interestingly, low deposition fluxes are observed during the summer season in Taipei (Lee et al., 2015; Huh et al., 2006) coinciding with high precipitation. This feature is well-captured in the model. Taipei has a typhoon season in summer when strong precipitation can occur in a very short period. The atmospheric $^7$Be could be removed quickly at the early stage of the precipitation event while at the later stage there is little $^7$Be left in the air that can be removed (Ioannidou and Papastefanou, 2006).

To examine the ability of model to simulate $^{10}$Be in polar regions, we compare model results with two sub-annual ice cores records (Fig. 9): the GRIP record from Greenland (1986-1990) (Heikkilä et al., 2008c) and the DSS record from Antarctica (2000-2009) (Pedro et al., 2011a). It should be noted that the direct measurements from ice cores are concentrations in the ice (atoms/g). To calculate deposition fluxes, the ice concentrations are multiplied with ice accumulation rates. However, for sub-annual accumulations, this bears large uncertainties. Therefore, we calculate the modeled $^{10}$Be concentrations for the selected sites using the model deposition fluxes at the selected sites timed by ice density and then divided by the corresponding model precipitation rates.

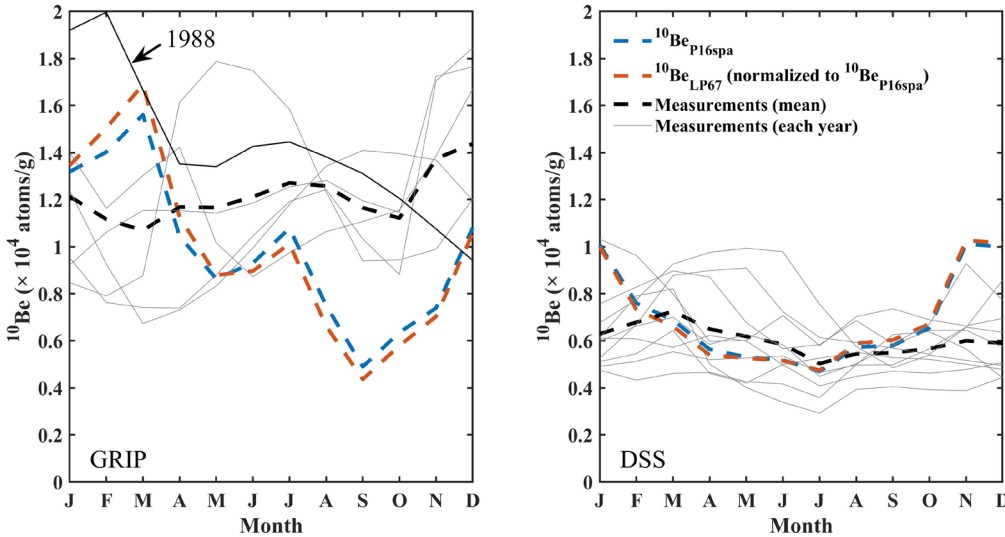

**Figure 9**. Seasonal cycle of simulated $^{10}$Be deposition fluxes (2008-2018) and measured $^{10}$Be deposition fluxes in GRIP (1986-1990) and DSS (2000-2009) ice cores. The solid lines (grey) refer to seasonal variations of the measurements for each year. The black solid line indicates seasonal data of measurements in the year 1988. The dashed lines indicate the averaged seasonal variations of measured $^{10}$Be (black), $^{10}$Be$_{P16spa}$ (blue), and $^{10}$Be$_{LP67}$ (red) concentrations.

Firstly, there is no consistent seasonal cycle in the GRIP $^{10}$Be measurement, indicating a strong role of local meteorology. The model does not reproduce the mean seasonal cycle partly because the model was not run for the exact same period. However, we note that the measurements for the year 1988 show an annual cycle similar to that in the model, suggesting that the model $^{10}$Be seasonality falls within the range of the observations. For the DSS site, the model simulates the austral winter minima but not the austral fall maxima (February-April). These model biases could be due to the limited model resolution and local effects (e.g., ice redistribution due to wind blow) that are not resolved by the model. Such discrepancies were also reported by previous model studies using the ECHAM5-HAM general circulation model ($2.8° × 2.8°$) over the overlap period (Heikkilä et al., 2008c; Pedro

et al., 2011b). Global model simulations at higher resolutions or using a regional model could help improve the agreements between model results and measurements at Greenland and Antarctica. However, it should be kept in mind that local surface processes can cause a high degree of spatial variability in the impurity concentrations in ice cores even on short distances (Gfeller et al., 2014), which cannot be resolved in climate models.

### 3.7 $^{10}$Be/$^{7}$Be ratio

Figure 10 shows the modeled zonal mean $^{10}$Be$_{P16spa}$/$^{7}$Be$_{P16spa}$ ratios during boreal spring (March-May) and austral spring (September-November), respectively, when the stratosphere-troposphere exchange is strong in either of the two hemispheres. Also shown are the comparison of the altitudinal profile of the $^{10}$Be$_{P16spa}$/$^{7}$Be$_{P16spa}$ ratio with measurements from three aircraft missions (Jordan et al., 2003). The model $^{10}$Be$_{P16spa}$/$^{7}$Be$_{P16spa}$ ratio generally lies within the ranges of measurements (Fig. 10c). Due to the decay of $^{7}$Be and long residence time in the stratosphere, the $^{10}$Be/$^{7}$Be ratio is higher (>1.5) in the stratosphere and increase over the altitude, with a maximum (>10) in the tropical stratosphere. During the period without strong stratospheric intrusion (e.g., autumn season in Northern Hemisphere, Fig.10b), the monthly $^{10}$Be/$^{7}$Be ratio near the surface is around 0.9~1. This surface $^{10}$Be/$^{7}$Be ratio could be up to 1.4 when the strong stratosphere-troposphere exchange happens (e.g., spring season in Northern Hemisphere, Fig. 10a).

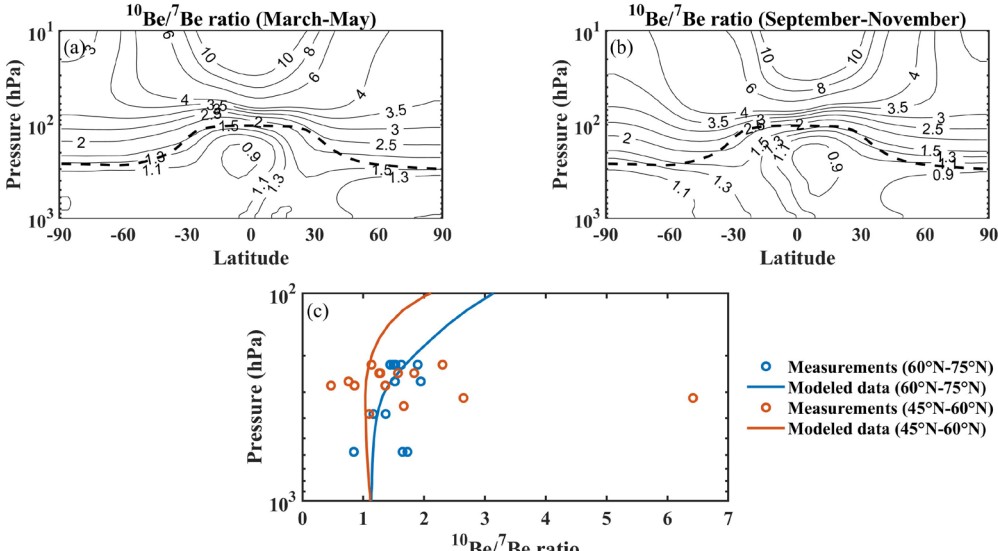

**Figure 10**. Upper panels: simulated $^{10}$Be$_{P16spa}$/$^{7}$Be$_{P16spa}$ ratio in spring (March-May) (a) and autumn (September-November) (b) averaged over the years 2008-2018. Lower panel (c): comparison between the annual averaged model $^{10}$Be$_{P16spa}$/$^{7}$Be$_{P16spa}$ ratios (lines) and those from measurements (circles; Jordan et al., 2003). The comparison is shown for the latitude bands of 60°N-75°N and 45°N-60°N, respectively.

Figure 11 compares model surface air $^{7}$Be$_{P16spa}$ and $^{10}$Be$_{P16spa}$ concentrations and $^{10}$Be$_{P16spa}$/$^{7}$Be$_{P16spa}$ ratios with monthly mean observations in Tokyo (Yamagata et al., 2019) during the period of 2008-2014. Here we mainly focus on the relative variations, and $^{7}$Be and $^{10}$Be data are normalized. The model captures the observed variability in Tokyo well. The $^{7}$Be and $^{10}$Be show a peak in early spring (March-May) while the $^{10}$Be/$^{7}$Be ratio shows a wider peak over March-July. The summer minima of $^{7}$Be and $^{10}$Be are due to strong scavenging associated with the monsoon/typhoon season precipitation. While the $^{10}$Be/$^{7}$Be ratio is independent of precipitation scavenging, the peaks of $^{10}$Be/$^{7}$Be coincide well with the enhancements of stratospheric contribution in the model.

This indicates that the $^{10}Be/^{7}Be$ ratio is a better indicator of the vertical transport and stratospheric intrusion influences than either tracer alone.

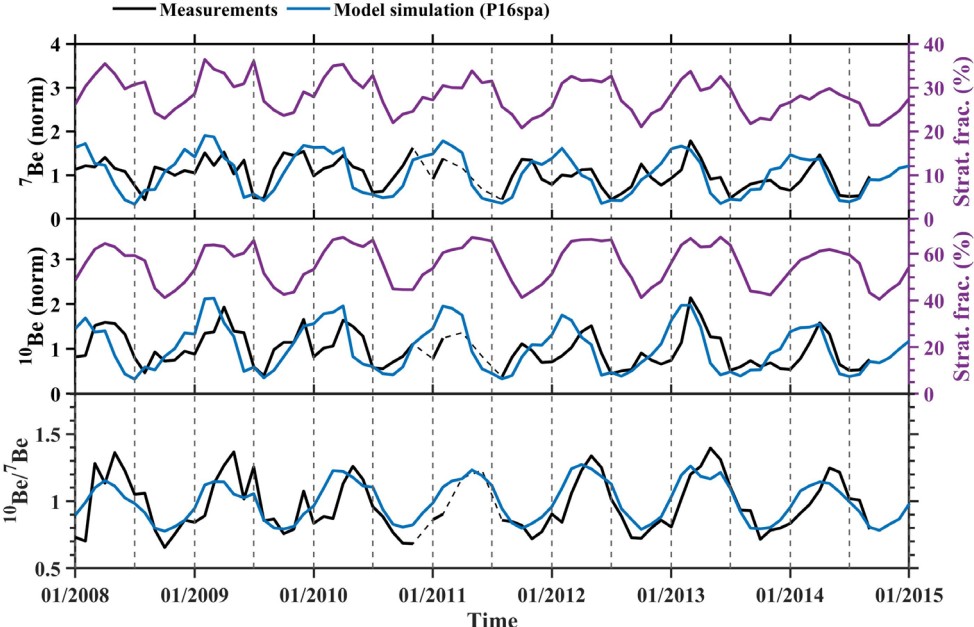

**Figure 11**. Comparison of monthly mean $^{7}Be$ (top panel), $^{10}Be$ (middle panel) concentrations, and $^{10}Be/^{7}Be$ ratio (bottom panel) between model results with P16spa production and measurements for the Tokyo station over the period 2008-2014. Noted that all $^{7}Be$ and $^{10}Be$ values are normalized to focus on variability. The dashed black line bridges the gap in measurements.

**3.8 Solar modulation influences**

Here we examine the ability of model to simulate the inter-annual variability of $^{7}Be$ surface air concentrations, especially whether the model can simulate the solar modulation influence using the updated production model. Figure 12 shows the comparison of model simulated annual mean surface air $^{7}Be$ concentrations with measurements during 2008-2018 from four sites: Kiruna, Ljungbyhed, Vienna and Hong Kong (Kong et al., 2022; Zheng et al., 2021a). The tropospheric $^{7}Be$ production rate from each site is also plotted for comparison as measured annual mean surface air $^{7}Be$ concentrations are predominantly influenced by the local tropospheric $^{7}Be$ production signal (Zheng et al., 2021a).

The model $^{7}Be_{P16spa}$ surface air concentrations show a better agreement with annual $^{7}Be$ measurements (higher R-value) compared to $^{7}Be_{LP67}$ concentrations at all surface sites (Fig. 12). The variability in the measurements (Kiruna, Ljungbyhed, and Vienna) agrees well with the trend in production, suggesting a dominant influence of solar modulations during this period. This is further supported by strong deviations between $^{7}Be_{P16spa}$ and $^{7}Be_{LP67}$ as no solar influence is considered in $^{7}Be_{LP67}$. This also emphasizes the importance of including solar modulation of the $^{7}Be$ and $^{10}Be$ production in modeling studies, especially for high-latitude regions. The mismatch of measurements and production at Kiruna from 2012 to 2015, together with the similar year-to-year variability between $^{7}Be_{P16spa}$ and $^{7}Be_{LP67}$, suggests the meteorological influence is dominant at Kiruna for this period. This also suggests that meteorological influences can suppress the solar signal in the $^{7}Be$ and $^{10}Be$ observations.

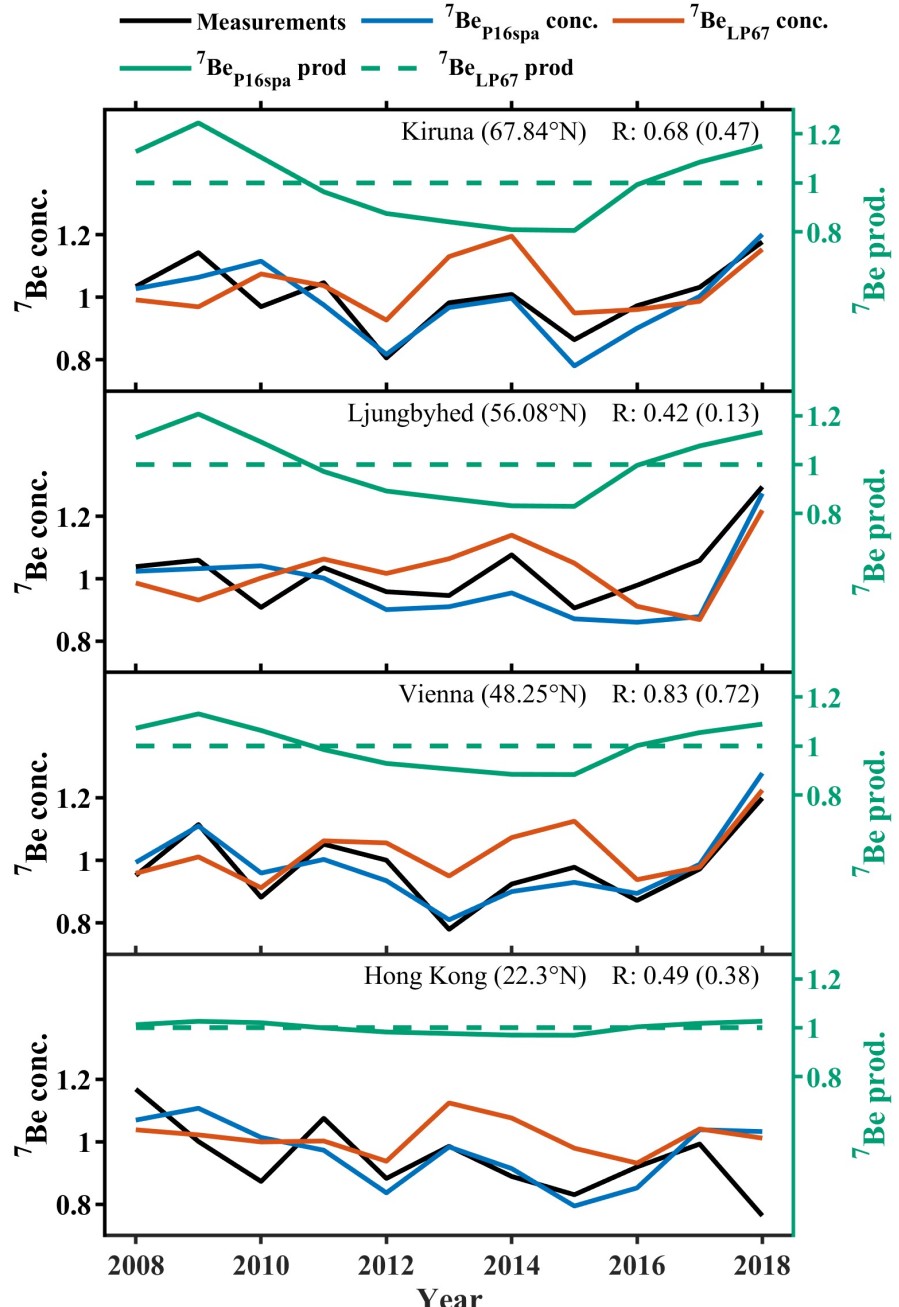

**Figure 12**. Comparison of annual mean model surface air [7]Be concentrations with measurements from 2008-2018. Also shown are the model tropospheric [7]Be production (green lines) at each station. All data are normalized by being divided by the mean over the first five years. The linear spearman correlation coefficient R-value is between [7]Be$_{P16spa}$ and measurements while the value in the bracket is between [7]Be$_{LP67}$ and measurements.

## 4 Summary and conclusions

We have incorporated the [7]Be and [10]Be production rates derived from the CRAC:Be model considering realistic spatial geomagnetic cut-off rigidities (P16spa) into the GEOS-Chem global chemical transport model, enabling the model output to be quantitatively comparable with the measurements. In addition to the standard simulation using P16spa production rate, we further conducted two sensitivity simulations: one with the default production rate in GEOS-Chem based on an empirical approach (LP67), and one with production rate from the CRAC:Be but

considering only geomagnetic cut-off rigidities for a geocentric axial dipole (P16). On global average, the LP67
production rate is 60% higher compared to those of P16 and P16spa. The P16 production rate shows some regional
differences (up to 50%) compared to the P16spa production rate.

In comparison with a large amount of air and deposition flux measurements, the model $^7\text{Be}_{P16spa}$ shows good
agreements with respect to surface air concentrations (93.7% of data within a factor of 2) and reasonably good
agreements regarding deposition fluxes (60.9% of data within a factor of 2). The model simulates well the surface
air concentration peaks in the subtropics associated strong downward transport from the stratosphere. This
agreement is better than those using the default production $^7\text{Be}_{LP16}$ and the $^7\text{Be}_{P16}$ production with simplified axis
symmetric dipole cut-off rigidity. The $^7\text{Be}_{LP67}$ simulation overestimates the absolute value of $^7\text{Be}$. The $^7\text{Be}_{P16}$
simulation tends to produce a positive bias (~18%) for the $^7\text{Be}$ deposition fluxes in East Asia region, nevertheless,
no large bias is found for $^7\text{Be}$ surface air concentrations. The surface deposition fluxes are more sensitive to the
production in the mid- and upper-troposphere and downward transport of $^7\text{Be}$ from the stratosphere, due to the
effect of precipitation scavenging throughout the troposphere.

For the first time, the ability of GEOS-Chem to simulate $^{10}\text{Be}$ is assessed with measurements. The model
$^{10}\text{Be}_{P16spa}$ results agree well with $^{10}\text{Be}$ observational data that were evaluated for dust influences or from the regions
less influenced by dust (e.g., polar regions), while underestimating most samples that were not corrected for dust
influences. This highlights the importance of examining the dust contribution to $^{10}\text{Be}$ measurements when using
these data to evaluate models.

Independent of the production models, surface $^7\text{Be}$ and $^{10}\text{Be}$ concentrations from all three simulations show
similar seasonal variations, suggesting a dominant meteorological influence. The model generally simulates well
the annual cycle of $^7\text{Be}$ surface air concentrations and deposition fluxes at most sites in terms of amplitude and
seasonality. The model fails to capture the "summer peak" in a few sites likely due to errors in convective transport
during summer.

The model $^{10}\text{Be}/^7\text{Be}$ ratios also lie within the measurements, suggesting the stratosphere-troposphere
exchange process is reasonably represented in the model. The mismatch of the peaks between $^7\text{Be}(^{10}\text{Be})$ and
$^{10}\text{Be}/^7\text{Be}$ ratios at the Tokyo site suggests that the $^{10}\text{Be}/^7\text{Be}$ ratio is a better indicator of the vertical transport and
stratospheric influences than either tracer alone as the ratio is independent of precipitation scavenging.

Finally, we demonstrate the value and importance of including time-varying solar modulation in $^7\text{Be}$ and
$^{10}\text{Be}$ production rates for model simulations of both tracers. It significantly improves the agreement of interannual
variations between the model and measurements, especially at those surface sites from mid- and high- latitudes.
The mismatch of trends in modeled $^7\text{Be}$ production rate and observed air concentrations at Kiruna from 2012-
2015 also suggests that the solar signal can be suppressed by meteorological influences.

In summary, we have shown that with the state-of-the-art P16spa production rate, the ability of GEOS-Chem
to reproduce the $^7\text{Be}$ and $^{10}\text{Be}$ measurements (including interannual variability of $^7\text{Be}$) is significantly improved.
While uncertainties in transport and deposition processes play a major role in the model performance, reduced
uncertainties in the production rates, as demonstrated in this study, allow us to use $^7\text{Be}$ and $^{10}\text{Be}$ tracers as better
tools for evaluating and testing transport and scavenging in global models. We recommend using the P16spa
(versus default LP67) production rate for GEOS-Chem simulations of $^7\text{Be}$ and $^{10}\text{Be}$ in the future.

*Author contributions*. MZ initiated the study. MZ performed the analysis and interpretation with contributions from HL and FA. MZ conducted the GEOS-Chem model simulations with the help from MW and ZL. All authors discussed the results and edited the manuscript.

*Competing interests*. The authors declare that there is no conflict of interest.

*Data and Code availability*. Observational data for model validation are available in the references described in section 2.3. The two compiled $^{10}$Be observation datasets are available in the Supplementary Information. The GEOS-Chem v14.0.2 model code, GEOS-Chem model output and $^7$Be and $^{10}$Be production rates are available at Zenodo repository (https://doi.org/10.5281/zenodo.8372652; Zheng et al., 2023a).

*Acknowledgments*. This project is supported by the Swedish Research Council (Dnr: 2021-06649) and the Swedish government funded Strategic Research Area: ModElling the Regional and Global Earth system, MERGE (MERGE). H. Liu acknowledges funding support from the NASA Modeling, Analysis and Prediction (MAP) program (grant 80NSSC17K0221) and Atmospheric Composition Campaign Data Analysis and Modeling program (grants NNX14AR07G and 80NSSC21K1455). F. Adolphi acknowledges support from the Helmholtz association (Grant number VH-NG 1501). R. Muscheler acknowledges support from the Swedish Research Council (grants DNR2013-8421 and DNR2018-05469). Z. Lu acknowledges Swedish Research Council Vetenskapsrådet (Grant No. 2022-03617). M. Wu acknowledges the National Natural Science Foundation of China (42111530184, 41901266). N. Prisle acknowledges the funding from the Academy of Finland (Grant Nos. 308238, 314175, and 335649). This project has received funding from the European Research Council (ERC) under the European Union's Horizon 2020 research and innovation programme, Project SURFACE (Grant Agreement No. 717022). The GEOS-Chem model is managed by the Atmospheric Chemistry Modeling Group at Harvard University. GEOS-Chem support team at Harvard University and Washington University in St. Louis (WashU) is acknowledged for their effort. GEOS-Chem input files were obtained from the GEOS-Chem Data Portal enabled by WashU.

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
