# Peer review of "Simulations of 7Be and 10Be with the GEOS-Chem global model v14.0.2 using state-of-the-art production rates"

_Geoscientific Model Development, 2023_

## Referee Comment (RC2)

**Review on:**

**Simulations of 7Be and 10Be with the GEOS-Chem global model v14.0.2 using state-of-the-art production rates by Zheng et al. (2023) submitted to GMD**

*Summary:*

*Zheng et al. are presenting an update of the GEOS-Chem global model, that aims for a more realistic representation of 7Be and 10Be production rates and with that a more realistic representation of the radionuclide transport and deposition. By comparing simulations with fixed production rates (represented by the year 1958 i.e., the year of the strongest solar maximum throughout the instrumental era) and experiments including time-varying solar modulation in the production calculation to observations, Zheng et al. clearly show the importance of the solar variability when trying to understand the measured near surface radionuclide content. According to their analysis, this is especially relevant for the middle and high latitudes.*

*The manuscript is generally very well structured, clear and easy to follow. The main statements are supported by the numerical experiments and the results are depicted by the aid of appealing figures.*

*I just have some minor comments and a few open technical questions (especially with respect to the aerosol model and the dry deposition and wet deposition ratios (please see below)) that should be addressed in a revised version of the manuscript. Afterwards, I recommend publication in GMD.*

*Minor comments:*

**L17: are useful aerosol tracers –** *The radionuclides 7Be and 10Be are actually not aerosols. They get attached to aerosols quickly and are then transported and deposited as such. The authors explain this later and I would remove "aerosol" here.*

**L48: removed by the wet deposition. –** *Not only wet deposition. Dry deposition, and sedimentation (see comment below) are also relevant deposition processes. Maybe "removed by different deposition processes" is more appropriate here.*

**L50: atmospheric atoms –** *Mostly oxygen and nitrogen, I guess. Please specify.*

**L69: and vertical transport –** *What about horizontal transport e.g., by storm-tracks in the middle latitudes?*

**L78-79: In comparison to other atmospheric models (e.g., Golubenko et al., 2021; Heikkilä et al., 2008b) –** *Spiegl et al. (2022) are using another approach based on EMAC and WASAVIES. I guess this paper deserves to be mentioned like the Sukhodolov et al. (2017).*

**L94: Previously –** *Maybe "In earlier studies".*

**L116-117: with a detailed description of stratospheric and tropospheric chemistry –** *Is a detailed chemistry code really relevant for the presented experiments? I guess the considered radionuclides may not be part of the chemistry scheme. If so, could the authors please comment on that?*

**L123-124: 7Be and 10Be are carried by ambient submicron aerosols after production and are removed by dry and wet deposition processes (Liu et al., 2001) -** *I think some more details, especially on the aerosol submodel, are essential here. Is the process of attachment explicitly modeled or did you treat the radionuclide loading physically like an aerosol directly? What distributions of aerosols did you use for the different domains (stratosphere vs. troposphere) with respect to diameter size and*

*distribution? Are volcanic effects included as well or is the size distribution constant over time? How is re-evaporation of aerosols treated in the model? I guess a profile (maybe to be included to the supplement) showing the considered mean diameter size as a function of height would be very helpful here. Also, is the process of sedimentation included in the experiments? While this process might be negligible for a small diameter size (stratospheric aerosol) it could become relevant for much bigger tropospheric aerosols. Could the authors please comment on this?*

**L127-128: Precipitation formation and evaporation fields from reanalysis data are used directly by the model wet deposition scheme. –** *If I understand correctly, the experiments are driven using a specified dynamics approach concerning 3d temperature and wind structure as well as 2d precipitation. What about the underlying SSTs. Could the authors comment on that?*

**L139: "stars" –** *Please explain more here for the readership that is not familiar with this terminology.*

**L140: multiplied with the mean production yield of 0.045 –** *Where does this number come from? Please specify.*

**L147-148: The P16 production model is regarded as the latest and most accurate production model for 7Be and 10Be –** *I believe that the approach presented here delivers accurate results. However, I do also think that alternative approaches could also deliver results that are just as convincing. Please rephrase to "one of the most accurate".*

**L180: 2012 to 2018 –** *Why was exactly this period chosen? It covers half of the solar cycle 24 and stops before the minimum of solar cycle 25. This is difficult to understand. Please explain.*

**L180: four-year spinup (2008-2011) –** *Please give some more details on the spinup period. Why was it necessary and how was it modelled (e.g., with respect to boundary conditions).*

**L186-187: also conducted for the year 2012 –** *Why "also"? It's the only experiment that has been conducted solely for the year 2012 in my understanding. Please clarify.*

**L188: done on a –** *"conducted using a…"*

**L193: abnormal value** – *"outliers"*

**L195-196: Usually, if the scatter plot of the model and measurements is within a factor of 2 of observations, we consider the model with reasonably good performance. –** *Please specify why you used this value (2) as a benchmark. Does it mean if the model results is 2 times larger than the observations you would consider it as reasonably good performance?*

**L200: for surface air 7Be concentrations** – *Please explain how the near surface air concentrations from the model integrations have been calculated before comparing them to observations. Did the authors use the closest box to the surface? I guess this information would be necessary.*

**Results:**

***General comment:*** *From here on a different font has been used and the figures and the manuscript becomes rather blurry. Is there a reason for this?*

***L231-238:*** *If I understand correctly, the main differences between the LP67 and P16 models are a result of the different state of the Sun (1958=solar max vs. 2012=between solar min and max), while the general physics of both models are rather similar. Is this correct? Can you explain why you used the year 2012 as a reference? Did you like to capture the mean state of the Sun? Please leave a comment in the text.*

**Figure2:** *The figure appears to be blurry. The axis and colorbarn labels should be bigger.*

**L277-279: … to capture local weather conditions –** *I agree especially e.g., the deposition fluxes over Greenland and Antarctica could be highly influenced by the model resolution used to capture the complex terrain (see e.g., Spiegl et al. (2022)). I think a comparison to individual stations at very unique locations is only possible using regional climate models. Also, the process of tropopause foldings could be influenced by the rather course resolution here.*

**Figure 3:** *What is actually shown here? Are these all the station data plotted against the model data? If so, did you use the closest grid box to a station for campariosn? Is this the annual mean surface air concentrations?*

**Figure 4 and 6:** *Axis labels are too small to read.*

**Section 3.2 and 3.3:** The authors nicely compare the model fluxes to the observations here and I don't have any comments. I just have one wish! Would the authors please be so kind and provide some map plots (maybe in the SI) that show the individual contributions of wet and dry deposition and (if applicable) sedimentation. I know from my own experience that getting a "correct" pattern is not an easy task. While the total fluxes may agree between different models, the individual contributions of different deposition processes (wet, dry and sedi) can be very different. E.g., compare Field et al. (2006), Heikkilä et al. (2013) and Spiegl et al. (2022). I do think the differences in the pattern are a result of different aerosol models and thus tracer treatment. This is why I would like to ask the authors to provide more details on this (see above). Seasonal map plots would be perfect, annual-mean sufficient. Thanks!

**L376: increasing trend** *– What is meant by increasing trend here?*

**Figure 7:** *This figure is blurry again and the axis and legend labels are too small to read. Please provide more information the caption on the data. Is it annual mean? Which period?*

**Page 14:** *Clear text again! From 15 on again blurry and different font.*

**L421: 2012-2018 while the measurements are based on the data availability during 2001-2015.** *– I think this is OK, since you only like to compare the general pattern.*

**Figure 8 and continued:** *This is a nice figure, but it needs some revision. The axis labels are to small, the grey text is difficult to read as well as the legend. And its blurry again.*

**On the "modelled stratospheric contribution"** *– Would the authors please specify how this was computed.*

**Figure 9:** *The grey text is difficult to notice.*

**L481: GIRP** *– GRIP*

**L481-488:** *I agree that the seasonal cycle is not well reproduced by the model with respect to polar latitudes and to some degree this could be related to the modelled period. However, I do think that the model resolution is just not high enough. Please see my comment on regional models above. Maybe this could be mentioned as a future perspective?*

---

## Author Comment (AC1)

We would like to thank both reviewers for their insightful and helpful comments. Below we reply to each comment point by point, showing the reviewers' comments in black and our responses in blue. Changes to the original manuscript are highlighted in **bold blue**. Note that the line numbers in the response are updated based on the revised manuscript.

Reviewer #1

This study discusses the use of cosmogenic isotopes $^7$Be and $^{10}$Be as aerosol tracers for studying atmospheric transport. By combining measurements of these isotopes with an atmospheric transport model, it is possible to gain insights into their transport and deposition processes and evaluate the model's performance. The study examines different production scenarios using the GEOS-Chem model driven by the MERRA-2 reanalysis and compares them to a large number of measurements. The results show that simulations considering realistic spatial distributions of geomagnetic cut-off rigidities (P16$_{spa}$) can accurately reproduce the surface concentrations, deposition fluxes, and vertical profiles of $^7$Be and $^{10}$Be. Simulations with the default production rate (LP67) tend to overestimate concentrations. The study also highlights the importance of including time-varying solar modulation in production calculations, which greatly improves the agreement between model results and measurements, particularly at mid- and high-latitudes. Overall, this research contributes to our understanding of aerosol transport and the role of cosmogenic radionuclides in atmospheric studies.

This work covers a highly relevant and rapidly evolving topic, but I am not sure that it fully aligns with the scope of the journal, since this study involves neither the development of a model nor a specific module of a full model but rather utilizes pre-existing code with only the input data being modified.

Response: We thank the reviewer for this comment. We believe this manuscript is within the journal scope as it is submitted under the manuscript type "Model evaluations". It is the first time that the ability of the GEOS-Chem model to simulate $^7$Be and $^{10}$Be is assessed against measurements using proper production rates of the two radionuclides. The model simulated $^7$Be was previously evaluated but the default production rate (LP67) used is rather outdated and does not consider changes in solar modulation. Even though a scale factor was applied to correct this solar modulation influence in previous studies, it is not ideal as the influence of the varying solar modulation dependents on latitude and altitude. Also, the GEOS-Chem simulations of $^{10}$Be have never been reported in the literature prior to the present study.

Additionally, in my opinion, the study requires significant improvement. Currently, the main shortcoming is related to the too-short modeling period (one cannot track the solar cycle over the 6-year period studied here). Furthermore, one of the scenarios (LP67) makes little sense since the model uses the outdated production model and the constant modulation potential for different levels of solar activity. However, there are several modern models available that account for changes in the modulation potential. Introducing a scaling coefficient is not correct in this context, especially considering that the authors are addressing the tiny regional

differences in the other two scenarios (P16 and P16$_{spa}$). Therefore, my main suggestion is **to add in the abstract and conclusion a strong recommendation not to use LP67 in future studies** (since on the global average, the LP67 production rate is 67% higher compared to those of P16 and P16$_{spa}$). **I also suggest extending the time range for P16 and P16$_{spa}$ modelling to at least 12 years, which is a minimum duration needed to estimate whether the model reproduces the solar cycle.** The inclusion of seasonal activity in the study is important, and the authors have addressed this aspect well. It would be beneficial to see results on a longer temporal scale as well. In my opinion, the main emphasis in the article could be placed on the fact that, for the first time, the ability of GEOS-Chem to simulate $^{10}$Be has been assessed using measurements considering a proper production model is used. Indeed, this is a significant improvement for the model.

Response: We thank the reviewer for these suggestions. The two main concerns are addressed as follows.

1). *The modeling period is too short and should cover one solar cycle.*

We now have rerun our model simulations for the period of 2002-2018 with the first six years as spin-up to make sure that $^{10}$Be reaches equilibrium in the stratosphere and the rest (2008-2018, 11 years) for analysis. Now, the simulations cover a whole solar cycle as the reviewer suggested. All results have been updated accordingly based on updated simulations. We would like to emphasize that our main conclusions have not changed based on the updated simulations.

2). *LP67 production should not be recommended (in the abstract and conclusion) to be used in future studies.*

We agree with the reviewer that the LP67 is outdated for the $^7$Be and $^{10}$Be simulations, since it does not consider the solar modulation influence and results in an offset compared to measurements. Actually, one of the main points of this manuscript is to use the GEOS-Chem model with more accurate and adequate production rate inputs (e.g., P16spa) for $^7$Be and $^{10}$Be simulations in the future. We also highlight in the abstract and in the conclusion that the LP67 production rate is not recommended.

Line 41-42 in the abstract: **"For future GEOS-Chem simulations of $^7$Be and $^{10}$Be, we recommend using the P16spa (versus default LP67) production rate".**

Line 608-609 in the conclusion: **"We recommend using the P16spa (versus default LP67) production rate for GEOS-Chem simulations of $^7$Be and $^{10}$Be in the future."**

We think it is necessary to keep LP67 scenario in this study. Both LP67 and P16 scenarios are sensitivity tests compared to the standard scenario P16spa. They allow us to evaluate how model simulations are biased using the LP67 and P16 production rates. The LP67 production

rate has previously been used in global three-dimensional chemical tracer models or general circulation models (e.g., GEOS-Chem and GISS GCM) (Brattich et al., 2017; Koch and Rind, 1998; Koch et al., 1996; Liu et al., 2016; Liu et al., 2001). For the LP67 scenario, we follow the previous studies (Koch et al., 1996; Liu et al., 2016) using a scale factor (1.39) to correct the solar modulation influence from the solar maximum year to a year with an average solar activity. Therefore, including the LP67 production rate in the present study also allows for the comparison of our work with those in the literature. We have rewritten the manuscript to clarify this point.

**Line 21-26 in the abstract**

**"Here we use the GEOS-Chem chemical transport model driven by the MERRA-2 reanalysis to simulate $^{7}$Be and $^{10}$Be with the state-of-the-art production rate from the CRAC:Be (Cosmic Ray Atmospheric Cascade: Beryllium) model considering realistic spatial geomagnetic cut-off rigidities (denoted as P16spa). We also perform two sensitivity simulations: one with the default production rate in GEOS-Chem based on an empirical approach (denoted as LP67), and the other with production rates from the CRAC:Be but considering only geomagnetic cut-off rigidities for a geocentric axial dipole (denoted as P16)."**

**Line 201-213**

**"The simulation with the P16spa production rate is considered as the standard simulation while the simulations with the P16 and LP67 production rates are sensitivity tests. The simulation with the P16 production rate is conducted to evaluate the influence of a simplified approximation of cutoff rigidities resulting from a geocentric dipole. In earlier studies, the LP67 production rate was used for global model simulations of $^{7}$Be (e.g., Brattich et al., 2017; Koch et al., 1996; Liu et al., 2016; Liu et al., 2001). The purpose of performing the simulation with the LP67 production rate is to evaluate to what extent model simulations are biased when applying the default LP67 production. Since the LP67 production rate applies only for the year 1958 (with a solar modulation function of about 1200 MeV) and does not consider the influences of the solar variations (e.g., 11-year solar cycle), it underestimates the production rate for the period of 2008-2018 that has an average solar modulation function of 500 MeV. To correct for this solar modulation influence, we follow the previous studies (e.g., Koch et al., 1996; Liu et al., 2016) by multiplying the model results by a scale factor of 1.39. It should be noted that this correction is not ideal as the effects of a varying solar modulation on cosmogenic radionuclide production rate depend on altitude and latitude."**

**Line 30:** It is crucial to highlight in the abstract the percentage of mismatch and provide a recommendation to readers not to use this scenario for future research, as it is unsuitable.

Response: Now it is included in the abstract (line 41-42).

**"For future GEOS-Chem simulations of $^7$Be and $^{10}$Be, we recommend using the P16spa (versus default LP67) production rate".**

**Line 50:** "Incoming galactic cosmic rays (GCRs)" not only galactic, but also solar cosmic rays too.

Response: We have changed it to "incoming cosmic rays".

Line 51-53: **"$^7$Be and $^{10}$Be are produced through interactions between atmospheric atoms (mostly oxygen and nitrogen) and incoming cosmic rays in the atmosphere (Lal and Peters, 1967, referred to as LP67 hereafter; Poluianov et al., 2016, referred to as P16 hereafter)."**

**Line 54-55:** as a reference for stratosphere/troposphere production distribution you can also use Heikkilä et al., 2013 (doi:10.1002/jgrd.50217) and Golubenko et al., 2022 (doi:10.1029/2022JD036726).

Response: Thanks for the references. Both have been added.

**Line 56:** also, Delaygue et al., 2015 (doi:10.3402/tellusb.v67.28582) discussed how the size of aerosol on beryllium transport.

Response: The reference has been added.

**Line 58:** Please, add approximately atmospheric residence times.

Response: The atmospheric residence time of $^{10}$Be (about 1-2 years) has been added to the sentence.

Line 58-60: **"$^{10}$Be has a half-life of 1.39 million years (Chmeleff et al., 2010) and its decay is thus negligible compared to its average atmospheric residence time (about 1-2 years) (Heikkilä et al., 2008)"**

**Line 76-78:** Please, also describe the accuracy of this model and the percentage of disagreement between simulation and real data.

Response: Thank you for the suggestion. We have revised the text to line 80-85:

**"By comparing the measurements with GEOS-Chem simulations over January-March 2003, Brattich et al. (2021) found that increased [7]Be values in surface air samples in Northern Europe in early 2003 were associated with the instability of the Arctic polar vortex. They also showed that, while the model generally simulates well the month-to-month variation in surface [7]Be concentrations, it tends to underestimate the observations (see their Table 2) partly due to the use of the default LP67 production rate for a solar maximum year (1958) in the GEOS-Chem model (Liu et al., 2001)."**

**Line 80-81:** It is incorrect to state that the model described in this sentence (e.g., CCM SOCOL, EMAC) can only be used as a free-running model. Models of this type can be employed as free-running for periods when reanalysis data is unavailable, they can be used with nudging for periods with reanalysis data. Nudging refers to the utilization of reanalysis data and provides similar capabilities to GEOS-Chem (with the only difference being the choice of reanalyses, such as Merra, ERA, or another dataset). On the other hand, GEOS-Chem is unable to function without reanalysis data, particularly when investigating the transport of cosmogenic isotopes in the past (e.g. T. Spiegl et. al., 2022 doi: 10.1029/2021JD035658). Free-run models, however, can be used in such cases. It's important not to mislead readers. In the articles referenced, the authors employ CCM SOCOL in combination with ERA-Interim.

Response: Thank you for pointing it out. To avoid confusion, we have removed the text in comparison with other models. We would like to mention that GEOS-Chem can be driven by not only reanalysis data but also output from general circulation models (e.g., Murray et al., 2021). Hence, it could also be used to investigate the transport of cosmogenic radionuclides in the past or future scenarios.

**Line 91:** The units of the solar modulation potential is MV, not MeV

Response: Thank you for the comment. Here we refer to the solar modulation function ($\Phi$, MeV), not the solar modulation parameter ($\phi$, MV). We have rewritten the sentence to avoid misunderstandings (line 91).

**"….a year with a high solar modulation function (i.e., high solar activity) of 1200 MeV (Herbst et al., 2017)."**

**Line 93-94:** Can you please clarify and rewrite this sentence: "Certain modifications of solar modulation need to be applied in simulations for different years to account for changes in solar modulation".

Response: Actually, this sentence is not necessary. We have therefore deleted it to avoid confusion.

**Line 100-104:** It would look more readable if each scenario is formatted as an item. For example:

- "Scenario I: GEOS-Chem default production using an empirical proximation (LP67 production);
- Scenario II: production derived from the "CRAC: Be" model considering realistic geomagnetic cut-off rigidity;
- Scenario III: a production derived from the "CRAC: Be" model considering only the dipole-moment of the geomagnetic field and an approximation of the resulting latitudinal variations in the cut-off rigidity (the so-called "Stoermer" cut-off)."

Or maybe it will be clearer if the text description here is changed to an overview of the performed simulations (Table 1 from the supplementary materials).

Response: Thank you for these suggestions. We have improved the paragraph following the reviewer's suggestion (line 99-109). The revised paragraph reads as

**"In this study, we incorporate global $^7$Be and $^{10}$Be production rates from the recently published "CRAC:Be" (Cosmic Ray Atmospheric Cascade: Beryllium) model (Poluianov et al., 2016) into the GEOS-Chem model. We simulate $^7$Be and $^{10}$Be using GEOS-Chem with the following three production scenarios.**

- **Scenario I: production rate derived from the "CRAC:Be" model considering realistic geomagnetic cut-off rigidity (P16spa production rate)**
- **Scenario II: production rate derived from the "CRAC:Be" model considering an approximation of geomagnetic cut-off rigidities using a geocentric axial dipole (P16 production rate)**
- **Scenario III: default production rate in GEOS-Chem using an empirical approximation (LP67 production rate)**

**Scenario I is treated as the standard simulation while the other two are sensitivity tests that also enable comparison to earlier studies. "**

**Line 115:** The model description lacks some details about the Beryllium module. As this journal is dedicated to models and their development, it is desired to have more specific information. Currently, there are many references, but there are no concise conclusions.

Response: Thank you for the suggestion. We have rewritten the model description section with more details about the representation of Beryllium properties and processes:

Line 120-154

"GEOS-Chem is a global 3-D chemical transport model (http://www.geos-chem.org) that simulates trace gases and aerosols in both the troposphere and stratosphere (Bey et al., 2001; Eastham et al., 2014). It is driven by archived meteorological data. We use version 14.0.2 (https://wiki.seas.harvard.edu/geos-chem/index.php/GEOS-Chem_14.0.2) to simulate the transport and deposition of atmospheric $^7$Be and $^{10}$Be. We drive the model with the Modern-Era Retrospective analysis for Research and Applications, Version 2 (MERRA-2) meteorological reanalysis (http://gmao.gsfc.nasa.gov/reanalysis/MERRA-2/; Gelaro et al., 2017a). MERRA-2 has a native resolution of 0.5° latitude by 0.667° longitude, with 72 vertical levels up to 0.01 hPa (80 km). Here the MERRA-2 data are re-gridded to 4° latitude by 5° longitude for input to GEOS-Chem for computational efficiency.

GEOS-Chem includes a radionuclide simulation option ($^{222}$Rn-$^{210}$Pb-$^7$Be-$^{10}$Be), which simulates transport (advection, convection, boundary layer mixing), deposition, and decay of the radionuclide tracers (e.g., Liu et al., 2001; Liu et al., 2004; Yu et al., 2018; Zhang et al., 2021) . The model uses the TPCORE algorithm of Lin and Rood (1996) for advection, archived convective mass fluxes to calculate convective transport (Wu et al., 2007), and the non-local scheme implemented by Lin and McElroy (2010) for boundary-layer mixing. As mentioned in the Introduction section, the standard GEOS-Chem model uses the LP67 beryllium production rate. After production, $^7$Be and $^{10}$Be attach to ambient submicron aerosols ubiquitously and their behavior becomes that of aerosols until they are removed by wet deposition (precipitation scavenging) and dry deposition processes. Note that neither is the process of attachment explicitly represented nor is the aerosol size distribution considered in the model. In addition, the decay process is included for the short-lived $^7$Be with a half-life time of 53.2-day. The decay is minor for the long-living $^{10}$Be, which has a half-life time of 1.38 million year (e.g., Chmeleff et al., 2010).

Wet deposition includes rainout (in-cloud scavenging) due to stratiform and anvil precipitation (Liu et al., 2001), scavenging in convective updrafts (Mari et al., 2000) , and washout (below-cloud scavenging) by precipitation (Wang et al., 2011). Scavenged aerosols from vertical layers above are allowed to be released to the atmosphere during re-evaporation of precipitation below cloud. In case of partial re-evaporation, we assume that half of the corresponding fraction of the scavenged aerosol mass is released at that level because some of the re-evaporation of precipitation are due to partial shrinking of the raindrops, which does not release aerosol (Liu et al., 2001) . MERRA-2 fields of precipitation formation and evaporation are used directly by the model wet deposition scheme. Dry deposition is based on the resistance-in-series scheme of Wesely (1989). The process of sedimentation is not included in the model.

**To quantify the stratospheric contribution to ⁷Be in the troposphere, we separately transport ⁷Be produced in the model layers above the MERRA-2 thermal tropopause (i.e., stratospheric ⁷Be tracer). This approach was previously used to study cross-tropopause transport of ⁷Be in GEOS-Chem (Brattich et al., 2021; Liu et al., 2001) and Global Modeling Initiative chemical transport models (Brattich et al., 2017; Liu et al., 2016). Stratospheric fraction of ⁷Be is defined as the ratio of the stratospheric ⁷Be tracer concentration to the ⁷Be concentration from the standard simulation."**

**Line 133-134:** please use the same number of decimal places in grid size.

Response: Thank you for the comment, but this expression of the grid size is standard (e.g., https://gmao.gsfc.nasa.gov/pubs/docs/Bosilovich785.pdf) and commonly used in the literature (e.g., Brattich et al., 2021; Gelaro et al., 2017b). Hence, we do not make changes.

**Line 134:** 80km à 80 km (space missing)

Response: Corrected.

**Line 155:** Please clarify is the energy spectrum of cosmic rays Ji is a function of the cutoff energy (Ec) or kinetic energy E?

Response: Thank you for the correction. It should be kinetic energy (E) instead of cutoff energy (Ec). We have corrected this in the manuscript (line 175-177):

**"The energy spectrum of cosmic rays $J_i$ is a function of the kinetic energy (E) and depends on the solar modulation function (Φ)(Herbst et al., 2017)"**

**Line 152:** Since the index «i» refers to different types of primary cosmic ray particles would you kindly provide information about the ratio of these particles? Do you consider particles that are heavier than alpha particles? If so, could you please explain how you incorporate them into your analysis?

Response: Poluianov et al. (2016) modelled the cosmogenic radionuclide production yield per incident proton and alpha particle of a given energy. Heavier particles can be considered as scaled (by nucleonic ratio) alpha particles. To obtain a cosmogenic radionuclide production rate, we assume a nucleonic ratio of alphas+heavies vs protons of 0.353 **(Koldobskiy et al., 2019)** in the local interstellar spectrum. We have added this information to the manuscript (line 172-174).

**"The *i* refers to different types of primary cosmic ray particles (e.g., proton, alpha and heavier particles). For modelling the contribution of alpha and heavier particles to the total production, their nucleonic ratio in the local interstellar spectrum was set to 0.353 (Koldobskiy et al., 2019)."**

**Line 170-173**: Please consider including a reference to the work by Nevalainen, Usoskin et al., 2013 (doi: 10.1016/j.asr.2013.02.020)? This study demonstrates that, on a global scale, there are no significant differences between the two scenarios mentioned – one using the Stoermer equation and the other utilizing the real geomagnetic cut-off rigidity inferred from particle trajectories. However, it does highlight minor discrepancies that may arise on a regional scale.

Response: The Stoermer cut-off rigidity we discussed here (expressed as only a function of the geomagnetic latitude and geomagnetic dipole moment; details see equation 5.8.2-2 in Beer et al., 2012) is a simpler eccentric dipole model (center dipole only) compared to the one discussed in Nevalainen et al. (2013). This model has already previously been shown to be deficient in low latitudes (Pilchowski et al., 2010). Even the more complex models used by Nevalainen et al. (2013) can significantly distort cut-off rigidities in some regions. Since the cut-off rigidity derived from simple eccentric dipole model has been used in several earlier studies (e.g., Koch and Rind, 1998; Koch et al., 2006; Koch et al., 1996; Liu et al., 2001), we would like to investigate the influence of using such a simple geomagnetic cut-off rigidity for $^{7}Be$ and $^{10}Be$ simulations.

We have added the sentence in the line (line 189-191).

**"Earlier studies suggested that using the simple centered dipole models (e.g., Stoermer cut-off rigidity) for cut-off rigidity approximation is limited as they can significantly distort the cut-off rigidity for some regions (e.g., low-latitude regions) (Nevalainen et al., 2013; Pilchowski et al., 2010)"**

**Line 183-184**: Please explain the rationale behind using LP67 if there is no possibility to use the modulation potential for years other than that of 1958. Initially, you focus on such a minor difference between P16 and P16spa suggesting that it may be important (cf. Nevalainen et al., 2013 doi:10.1016/j.asr.2013.02.020). However, a much stronger effect of the solar modulation potential is neglected in the LP67 scenario. This is unacceptable because 1958 was the year of the maximum activity, while the years 2012-2014, analyzed here, were during the period of reduced activity, and the assumption of scaling the solar potential is not physically plausible. Previous studies (already mentioned in the paper) have already demonstrated that LP67 has major inaccuracies, and it is better to use P16 or P16spa. If the GEOS-Chem model does not allow for changing the solar modulation potential, it is a serious drawback that raises doubts about the entire experiment 1. Based on the above, you could perform all three scenarios for

the year 1958 (although this may have little sense, as the comparison between LP67 and P16 or P16spa has already been done earlier).

Response: The LP67 scenario is included as a sensitivity test to investigate to what extent the model results will be biased if the LP67 production rate is used considering that the LP67 production rate has been used in global model simulations in earlier studies (Brattich et al., 2017; Brattich et al., 2021; Liu et al., 2016; Liu et al., 2001). The scale factor follows previous studies (e.g., Koch et al., 1996; Liu et al., 2016) to adjust the LP67 production from a solar maximum year (1200 MeV) to a year with an average solar activity (500 MeV). Indeed, this correction method used in previous studies is not ideal as the influence of the varying solar modulation is latitudinally and vertically dependent (already denoted in line 212-213). We agree with the reviewer that P16spa should be recommended for use given the disadvantage of the LP67 production rate. We have highlighted in both the abstract and the conclusions our recommendation of using the P16spa production for GEOS-Chem simulations of $^7$Be and $^{10}$Be in future studies.

**Line 203-209**: Again, as in lines 100-104. For readers, it's not clear when all options look like one sentence. Please use items.

Response: We have itemized the data sources as suggested (line 233-241).

**It includes the data from:**
- **The Environmental Measurements Laboratory (EML, https://www.wipp.energy.gov/namp/emllegacy/index.htm) Surface Air Sampling Program (SASP), which began in the 1980s,**
- **The ongoing international monitor program Radioactivity Environmental Monitoring (REM) network (e.g., Hernandez-Ceballos et al., 2015; Sangiorgi et al., 2019),**
- **International Monitoring System (IMS) organized by the Comprehensive Nuclear-Test-Ban Treaty Organization (CTBTO) (e.g., Terzi and Kalinowski, 2017),**
- **Some additional datasets in publications not included in the above programs.**

**Line 220**: Please, add more details about $^{10}$Be surface air measurements used in this study.

Response: We added some details regarding the surface air measurements (line 256-259).

**"The air samples are continuously collected by filters using a high-flow aerosol sampler. The sampling volume was approximately 700 m³ of air for daily samples (e.g., Liu et al., 2022) and between 3000 m³ and 5000 m³ for weekly samples (e.g., Yamagata et al., 2019)."**

**Line 231-253:** I believe that Figure 1 and the discussion are unnecessary here, as already mentioned by the authors in the introduction, the advantage of P16 over LP67 has been demonstrated in previous studies. It does not provide any practical utility in this context since the GEOS-Chem model with input parameters from LP67 cannot be applied to any other period except for the year 1958. However, the authors here study the period of 2012-2014, when the solar modulation potential was almost half that of 1958.

Response: We have shortened the discussion to avoid unnecessary repetition and moved Figure 1 to the supplementary (Figure S2).  However, we think it is necessary to show the LP67 and P16 production distribution and the differences between them. Here we also show differences between LP67 and P16 production rates over the stratosphere and troposphere, which were not discussed in detail in previous studies.

We agree with the reviewer that a comparison between LP67 in 1958 and P16 over the period of 2008-2018 is unnecessary since they have different solar modulation functions. We have removed this part from the discussion and the figure.

**Line 254-263:** I suggest discussing not only the presence of anomalies but also their underlying causes in more detail, based on earlier studies as proposed by Neväläinen et al. (2013), and searching for similar cases.

Response: This is already discussed in section 2.2. The underlying cause of differences between the P16 and P16spa production rates is due to the differences between real cut-off rigidity and the approximation of geomagnetic cut-off rigidity using a centered dipole moment. We have included Nevalainen et al. (2013) in section 2.2 as the reference for the statement that such a simple approximation of geomagnetic cut-off rigidity (e.g., Stoermer cut-off rigidity) may not result in systematic errors on a global scale but could lead to significant errors on a regional scale.

**Figure 2:** The colour scale in Figure 2 (especially the upper panel) should be made clearer and possibly add contours to the plot. Additionally, please make the latitude numbering bold, similar to panel (d), to ensure consistency and clarity. The current presentation lacks cohesiveness.

Response: We have remade Figure 2 (now it is updated as Figure 1) to make it clearer and consistent.

[Figure]

**Figure 1**. Upper panels: Spatial distribution of (a) P16spa and (b) P16 $^7$Be production rates at 825 hPa over the period of 2008-2018. Lower panels: (c) Relative differences, i.e., ($^7$Be$_{P16spa}$-$^7$Be$_{P16}$)/ $^7$Be$_{P16}$ ×100%, between production rates with and without considering the detailed spatial cut-off rigidity. (d) Relative differences of the zonal mean production rates between P16spa and P16 at 30°N.

**Line 270-271:** I would like to emphasize once again that comparing the results of P16 and LP67 is not appropriate because we cannot use the same modulation potential for both 1958 and 2012, as done in the LP67 scenario.

Response: As explained in the responses above, the LP67 scenario is considered a sensitivity simulation as the LP67 production rate has been commonly used in global models including GEOS-Chem (e.g., Brattich et al., 2017; Koch and Rind, 1998; Koch et al., 1996; Liu et al., 2016; Liu et al., 2001). To correct the solar modulation influence from a solar maximum (1200 MeV) to the period of 2008-2018 with an average solar modulation function of 500 MeV, we follow the previous studies that applied a scale factor (1.39) to the model $^7$Be using LP67. This is now clearly stated in line 206-213.

**Line 314:** The use of scaling coefficients does not make sense because there are currently several models available that account for changes in modulation potential. How did you define

these coefficients, and what is its purpose? Once again, I suggest removing this scenario from the study.

Response: The scaling factor is only applied for the LP67 to correct the solar modulation influence as LP67 is only validated for the year 1958. This scale factor (1.39) has been used in earlier studies for LP67 production rates (e.g., Koch et al., 1996; Liu et al., 2016), to correct the solar modulation influence from a solar maximum to the period with an averaged solar activity (500 MeV). We agree with the reviewer that this correction method used in previous studies is not ideal as the influence of the varying solar modulation is latitudinally and vertically dependent (already denoted in line 212-213).

**Figure 4:** Could you please use a consistent color palette throughout the entire work? You chose a palette in Figure 1, and it would be convenient to use the same palette in the rest of the figures. Additionally, the measurements in the figure tend to blend with the background on the overall map. Could they be made slightly brighter (bolder)?

Response: We now made the color palette consistent for the concentration map by using only the color palette (light blue to dark blue). The measurement points are also made bolder.

**Line 327:** The contribution of the stratospheric and tropospheric fractions was described, in particular, by Heikkilä et al. 2009 (doi:10.5194/acp-9-515-2009), so this sentence only confirms her assessment.

Response: Thank you for the comment. However, Heikkilä et al. (2009) only looked at the stratospheric fraction in the $^{10}$Be deposition flux, not in the $^{10}$Be air concentrations. Here, we found that such stratospheric fraction is different in the air concentration compared to the deposition.

**Line 452**: If you normalize $^{7}$Be$_{LP67}$ and $^{10}$Be$_{LP67}$ to Be$_{P16spa}$, please indicate this in the legend of all the figures where a comparison with measurements is presented.

Response: We indicated this in the legend of figures when $^{10}$Be$_{LP67}$ is normalized to $^{10}$Be$_{P16spa}$ (Figure 9) or $^{7}$Be$_{LP67}$ is normalized to $^{7}$Be$_{P16spa}$ (Figure 7 and Figure 8)

Please make sure to give more attention in the conclusions to the fact that LP67, when compared to P16spa, provides distorted values.

Response: We have highlighted in the abstract and conclusions that we recommend using the P16spa production rate.

Reviewer #2
Review on:
Simulations of 7Be and 10Be with the GEOS-Chem global model v14.0.2 using state-of-the-art production rates by Zheng et al. (2023) submitted to GMD

Summary:
Zheng et al. are presenting an update of the GEOS-Chem global model, that aims for a more realistic representation of 7Be and 10Be production rates and with that a more realistic representation of the radionuclide transport and deposition. By comparing simulations with fixed production rates (represented by the year 1958 i.e., the year of the strongest solar maximum throughout the instrumental era) and experiments including time-varying solar modulation in the production calculation to observations, Zheng et al. clearly show the importance of the solar variability when trying to understand the measured near surface radionuclide content. According to their analysis, this is especially relevant for the middle and high latitudes.

The manuscript is generally very well structured, clear and easy to follow. The main statements are supported by the numerical experiments and the results are depicted by the aid of appealing figures. I just have some minor comments and a few open technical questions (especially concerning the aerosol model and the dry deposition and wet deposition ratios (please see below)) that should be addressed in a revised version of the manuscript. Afterwards, I recommend publication in GMD.

Response: We thank the reviewer for the very positive comments.

Minor comments:
L17: are useful aerosol tracers – The radionuclides 7Be and 10Be are actually not aerosols. They get attached to aerosols quickly and are then transported and deposited as such. The authors explain this later and I would remove "aerosol" here.

Response: The "aerosol" is removed as suggested.

L48: removed by the wet deposition. – Not only wet deposition. Dry deposition, and sedimentation (see comment below) are also relevant deposition processes. Maybe "removed by different deposition processes" is more appropriate here.

Response: We have changed the sentence to: **"…both $^7$Be and $^{10}$Be in the troposphere are mainly removed by wet deposition."**

L50: atmospheric atoms – Mostly oxygen and nitrogen, I guess. Please specify.

Response: Added accordingly:

**"$^7$Be and $^{10}$Be are produced through interactions between atmospheric atoms (mostly oxygen and nitrogen) and incoming cosmic rays…"**

L69: and vertical transport – What about horizontal transport e.g., by storm-tracks in the middle latitudes?

Response: It is now included in the text.

**Line 71-73: "…vertical transport in the troposphere (Ajtic et al., 2018; Aldahan et al., 2001; Zheng et al., 2021) and large-scale atmospheric circulations (Hernández-Ceballos et al., 2022; Terzi and Kalinowski, 2017)."**

L78-79: In comparison to other atmospheric models (e.g., Golubenko et al., 2021; Heikkilä et al., 2008b) – Spiegl et al. (2022) are using another approach based on EMAC and WASAVIES. I guess this paper deserves to be mentioned like the Sukhodolov et al. (2017).

Response: The sentence has been deleted in the revised text to avoid confusion for the readers as suggested by the reviewer #1.

L94: Previously – Maybe "In earlier studies".

Response: Changed accordingly.

L116-117: with a detailed description of stratospheric and tropospheric chemistry – Is a detailed chemistry code really relevant for the presented experiments? I guess the considered radionuclides may not be part of the chemistry scheme. If so, could the authors please comment on that?

Response: No, the chemistry code is not directly relevant for radionuclide simulations. We have changed the sentence.

**Line 120-121 "GEOS-Chem is global 3-D chemical transport model (http://www.geos-chem.org) that simulates trace gases and aerosols in both the troposphere and stratosphere (Eastham et al., 2014; Bey et al., 2001)."**

L123-124: 7Be and 10Be are carried by ambient submicron aerosols after production and are removed by dry and wet deposition processes (Liu et al., 2001) - I think some more details, especially on the aerosol submodel, are essential here. Is the process of attachment explicitly modeled or did you treat the radionuclide loading physically like an aerosol directly? What distributions of aerosols did you use for the different domains (stratosphere vs. troposphere) with respect to diameter size and distribution? Are volcanic effects included as well or is the size distribution constant over time? How is re-evaporation of aerosols treated in the model? I guess a profile (maybe to be included to the supplement) showing the considered mean diameter size as a function of height would be very helpful here. Also, is the process of sedimentation included in the experiments? While this process might be negligible for a small diameter size (stratospheric aerosol) it could become relevant for much bigger tropospheric aerosols. Could the authors please comment on this?

Response: Thank you for the suggestions. We have rewritten the model description section (section 2.1) with more details. Please refer to our response to reviewer #1's comments.

The gravitational sedimentation process is not included in our GEOS-Chem simulations of $^7$Be and $^{10}$Be. Earlier modeling studies have suggested that sedimentation does not play an important role in removing $^7$Be from the stratosphere because the transport and radioactive decay processes are faster (e.g., Delaygue et al., 2015; Golubenko et al., 2021). Sedimentation was also not included for $^7$Be and $^{10}$Be in some other global models (e.g., SOCOL; Golubenko et al., 2021; Sukhodolov et al., 2017).

The volcanic effects are not considered. However, we would like to mention that the volcanic influence on $^7$Be and $^{10}$Be deposition is still not well understood. In a recent study by Zheng et al. (2023), they did not find a strong increase in $^{10}$Be depositions in Greenland and Antarctica after the Pinatubo eruption in 1991. Future studies could focus on such volcanic influence and the mechanism behind it.

L127-128: Precipitation formation and evaporation fields from reanalysis data are used directly by the model wet deposition scheme. – If I understand correctly, the experiments are driven using a specified dynamics approach concerning 3d temperature and wind structure as well as 2d precipitation. What about the underlying SSTs. Could the authors comment on that?

Response: MERRA-2 fields for input to GEOS-Chem do not include sea surface temperature. But surface skin temperature is used in, for instance, the parameterization of sea salt emissions. It is not relevant to radionuclide simulations. No Changes are made.

L139: "stars" – Please explain more here for the readership that is not familiar with this terminology.

Response: This refers to nuclear disintegration due to the spallation of neutrons and protons with the oxygen and nitrogen atoms. We have rewritten the sentence to clarify this (line 157-159).

**"These production rates are calculated using an analytically estimated rate of nuclear disintegration (stars) in the atmosphere (stars/g air/s), multiplied by the mean production yield of 0.045 atoms/star for $^7$Be and 0.025 atoms/star for $^{10}$Be (Lal and Peters, 1967)."**

L140: multiplied with the mean production yield of 0.045 – Where does this number come from? Please specify.

Response: This number is from the study by Lal and Peters (1967), which is now cited in the sentence.

L147-148: The P16 production model is regarded as the latest and most accurate production model for 7Be and 10Be – I believe that the approach presented here delivers accurate results. However, I do also think that alternative approaches could also deliver results that are just as convincing. Please rephrase to "one of the most accurate".

Response: Corrected as suggested.

L180: 2012 to 2018 – Why was exactly this period chosen? It covers half of the solar cycle 24 and stops before the minimum of solar cycle 25. This is difficult to understand. Please explain.

Response: There is no special consideration regarding the choice of this period. We would like to mention that now we have rerun the simulation from 2002-2018 with the first 6 year as spin-up to allow $^{10}$Be to reach equilibrium in the atmosphere. The rest of the period (2008-2018) is used for analysis.

L180: four-year spinup (2008-2011) – Please give some more details on the spinup period. Why was it necessary and how was it modelled (e.g., with respect to boundary conditions).

Response: The spin-up is to allow $^{10}$Be to reach equilibrium in the atmosphere. This is now clearly mentioned in the manuscript. It should be mentioned that we now re-ran the model to have the simulations covering a whole solar cycle (2008-2018) and we have six years for the spin-up which follows earlier studies (e.g., Golubenko et al., 2021). We have started the GEOS-Chem model with an initial concentration of zero for $^7$Be and $^{10}$Be and let it run for 6 years for spin-up (because of the long stratospheric lifetime of $^{10}$Be). Then the period of 2008-2018 is used for analysis.

Line 213-215: **"All simulations are performed from 2002 to 2018 with the first six-year as spin-up to make sure the $^{10}$Be nearly reaches equilibrium in the atmosphere and the 2008-2018 period (11 years) for analysis."**

L186-187: also conducted for the year 2012 – Why "also"? It's the only experiment that has been conducted solely for the year 2012 in my understanding. Please clarify.

Response: There is no special consideration regarding the selection of the year 2012. Note that now we also run the simulation with the P16 production for the same period as the one with the P16spa production.

L188: done on a – "conducted using a…"

Response: Changed as suggested.

L193: abnormal value – "outliers"
Response: Changed as suggested.

L195-196: Usually, if the scatter plot of the model and measurements is within a factor of 2 of observations, we consider the model with reasonably good performance. – Please specify why you used this value (2) as a benchmark. Does it mean if the model results is 2 times larger than the observations you would consider it as reasonably good performance?

Response: This value has been used in many model-measurement comparison studies (Brattich et al., 2021; Heikkilä et al., 2008). The fraction of modeled concentrations within a factor of 2 of observations (FA2) is calculated, i.e., for which $0.5 < X_{model}/X_{observation} < 2$. This implies that the model values should be greater than half of the observations but less than doubled observations. We now have added those references in the sentences (line 222-224).

**"Usually, if the scatter plot of the model and measurements is within a factor of 2 of observations, the model is considered to have a reasonably good performance (e.g., Brattich et al., 2021; Heikkilä et al., 2008)."**

L200: for surface air 7Be concentrations – Please explain how the near surface air concentrations from the model integrations have been calculated before comparing them to observations. Did the authors use the closest box to the surface? I guess this information would be necessary.

Response: The model value selected for comparison is the one from the bottom grid box closest to the corresponding station. We now added this information in section 2.3 (line 224-226).

**"For model comparison with surface air concentrations, the model value from the bottom grid box closest to the corresponding measurement site is selected."**

Results:

General comment: From here on a different font has been used and the figures and the manuscript becomes rather blurry. Is there a reason for this?

Response: We are not sure why this happens. We checked the font in all figures to make sure the font name is consistently as "Times New Roman". We have inserted the higher-quality figures into the manuscript to make sure the figures are not blurry.

L231-238: If I understand correctly, the main differences between the LP67 and P16 models are a result of the different state of the Sun (1958=solar max vs. 2012=between solar min and max), while the general physics of both models are rather similar. Is this correct? Can you explain why you used the year 2012 as a reference? Did you like to capture the mean state of the Sun? Please leave a comment in the text.

Response: As stated in the Introduction, the LP67 production is calculated based on observations made during the solar maximum year of 1958 while the P16 is based on full Monte-Carlo simulations of the cosmic-ray-induced atmospheric nucleonic cascade (e.g., Masarik and Beer, 1999; Poluianov et al., 2016). Hence, the two models use different methods.

There are no special considerations regarding the use of the year 2012. Now we only compare the LP67 production rate with the P16 production rate for the same year 1958 in section 3.1 following reviewer #1's suggestion.

Figure2: The figure appears to be blurry. The axis and colorbarn labels should be bigger.

Response: We have made the axis and color bar labels bigger.

L277-279: … to capture local weather conditions – I agree especially e.g., the deposition fluxes over Greenland and Antarctica could be highly influenced by the model resolution used to capture the complex terrain (see e.g., Spiegl et al. (2022)). I think a comparison to individual stations at very unique locations is only possible using regional climate models. Also, the process of tropopause foldings could be influenced by the rather coarse resolution here.

Response: We added the reference Spiegl et al. (2022) and the "tropopause folding" that could be influenced by the model resolution in the sentence (line 307-310).

**"Further, the limited model resolution applied here may not be able to capture meteorological conditions on local scales (e.g., precipitation, convection, and tropopause**

**folding) in some sites (e.g., Spiegl et al., 2022; Yu et al., 2018), especially for coastal regions when the sub-grid scale orographic precipitation is important."**

Figure 3: What is actually shown here? Are these all the station data plotted against the model data? If so, did you use the closest grid box to a station for the campariosn? Is this the annual mean surface air concentrations?

Response: Yes, correct. Shown in the figure are the comparisons between model results (for the bottom grid box closest to the corresponding measurement site) and measurements. This is added in line 224-226 (see the reply above). The model results are averaged over the simulated period 2008-2018, while the measurements are based on the data availabilities. This is added in the caption of Figure 3.

**"Figure 3. Left column: (a) modeled $^7Be_{P16spa}$ surface air concentrations (mBq/m$^3$) and (b) deposition fluxes (Bq/m$^2$/yr) averaged over the period of 2008-2018. Color-coded dots denote $^7Be$ measurements. Right column: zonal mean of (c) observed $^7Be$ surface air concentrations and (d) deposition fluxes (black lines, for each 5° latitude bin) compared with the model simulation using the P16spa production rates (blue lines). Dots are individual measurements. The error bars indicate one standard deviation. The outliers, defined as more than three scaled median absolute deviations (MAD) away from the median, are excluded from the calculation. The observations are averaged over the years available."**

Figure 4 and 6: Axis labels are too small to read.

Response: We have increased the axis labels to make them more readable.

Section 3.2 and 3.3: The authors nicely compare the model fluxes to the observations here and I don't have any comments. I just have one wish! Would the authors please be so kind and provide some map plots (maybe in the SI) that show the individual contributions of wet and dry deposition and (if applicable) sedimentation. I know from my own experience that getting a "correct" pattern is not an easy task. While the total fluxes may agree between different models, the individual contributions of different deposition processes(wet, dry and sedi) can be very different. E.g., compare Field et al. (2006), Heikkilä et al. (2013) and Spiegl et al. (2022). I do think the differences in the pattern are a result of different aerosol models and thus tracer treatment. This is why I would like to ask the authors to provide more details on this (see above). Seasonal map plots would be perfect, annual-mean sufficient. Thanks!

Response: Thank you for the comment. The deposition patterns of $^7Be$ and $^{10}Be$ are mainly controlled by wet deposition which is related to precipitation. Below, we present a figure showing the fractions of $^{10}Be$ wet to total deposition fluxes during 2008-2018, as simulated by GEOS-Chem. The color bar follows Figure 3 in Heikkilä et al. (2011) for a better comparison.

[7]Be deposition fluxes (not shown) show similar patterns as [10]Be. GEOS-Chem simulates somewhat higher wet-to-total deposition flux ratios compared to those simulated by ECHAM5-HAM (Heikkilä et al., 2011) or ECHAM/MESSy (Spiegl et al., 2022).

While different contributions of wet and dry depositions are mainly due to the tracer treatment in different global models, it is difficult to determine which model better represents this "pattern" due to the lack of adequate measurements for validation, different periods of model simulations and different model resolutions. We have added this figure in the supplementary as Figure S7 and updated the manuscript accordingly.

**Line 428-430: "The wet deposition contributes to about 97% of total deposition for [7]Be and [10]Be (Table 1; Fig. S7), which is slightly higher than the ~93% contribution in previous model studies (Heikkilä et al., 2008; Koch et al., 1996; Spiegl et al., 2022)."**

[Figure]

**Fraction of [10]Be wet to total deposition**

**Figure S7. Fractions of wet to total deposition fluxes of [10]Be modeled by GEOS-Chem averaged over the period of 2008-2018.**

L376: increasing trend – What is meant by increasing trend here?

Response: We mean that the [7]Be concentrations at high latitudes (60°N-75°N; Figure 6a) increase with altitude (i.e., without a peak at 20 km like the ones at mid-latitudes). We have rewritten the sentence to make it clear (line 399-400).

**"The feature that [7]Be increases with altitude without a peak at 22 km at northern high latitudes (60°N-75°N) is also captured by the model (Fig. 6a)."**

Figure 7: This figure is blurry again and the axis and legend labels are too small to read. Please provide more information on the caption on the data. Is it annual mean? Which period?

Response: We have increased the font size of the axis and labels to make it clear. The model results are annually averaged over the period of 2008-2018. This information is added in the caption of Figure 7 (now Figure 6). The $^7$Be measurements are from the EML/HASP program which covers the period from 1962 to 1983, as mentioned in section 2.4. The figure caption now reads:

**"Figure 6. Comparison of the vertical profile between measurements (circles) and model zonal mean $^7$Be$_{p16spa}$ and $^{10}$Be$_{p16spa}$ concentrations for each latitudinal band (15°) over the period of 2008-2018. The $^7$Be (circle with error bar) observations (from the EML/HASP program) are averaged for the altitude band of every 2 km where more than 5 samples are available. We exclude the outlier from the calculation, which is defined as more than three scaled median absolute deviations (MAD) away from the median. The $^{10}$Be profile measurements are mainly taken from Dibb et al. (1994, 1992) and Jordan et al. (2003)."**
Page 14: Clear text again! From 15 on again blurry and in different font.

Response: We have reproduced the figures with a consistent font and high resolution.

L421: 2012-2018 while the measurements are based on the data availability during 2001-2015. – I think this is OK, since you only like to compare the general pattern.

Response: Thank you for the comment. The model simulations do not necessarily cover the same period due to the data availability.

Figure 8 and continued: This is a nice figure, but it needs some revision. The axis labels are to small, the grey text is difficult to read as well as the legend. And its blurry again.

Response: Thank you for the comment. We have increased the size of the text in the figure to make it bigger and more readable.

Figure 8, On the "modelled stratospheric contribution" – Would the authors please specify how this was computed.

Response: Thank you for bringing it up. We have added a short paragraph in the model description section (section 2.1) to specify how stratospheric contribution is quantified.

**Line 149-154: "To quantify the stratospheric contribution to $^7$Be in the troposphere, we separately transport $^7$Be produced in the model layers above the MERRA-2 thermal tropopause (i.e., stratospheric $^7$Be tracer). This approach was previously used to study cross-tropopause transport of $^7$Be in GEOS-Chem (Brattich et al., 2021; Liu et al., 2001) and Global Modeling Initiative chemical transport models (Brattich et al., 2017; Liu et**

**al., 2016). The Stratospheric fraction of $^7$Be is defined as the ratio of the stratospheric $^7$Be tracer concentration to the $^7$Be concentration from the standard simulation.”**

Figure 9: The grey text is difficult to notice.
Response: We have changed the color to dark grey.

L481: GIRP – GRIP

Response: Corrected. Thank you for noticing.

L481-488: I agree that the seasonal cycle is not well reproduced by the model with respect to polar latitudes and to some degree this could be related to the modelled period. However, I do think that the model resolution is just not high enough. Please see my comment on regional models above. Maybe this could be mentioned as a future perspective?

Response: Thank you for the suggestion. We have added a sentence in line 509-510 accordingly.

[revised manuscript text omitted]

---

## Author Response (AR2)

Dear Editor,

Thank you for your comments and suggestions. Please find below our responses in blue.

1. The first refers to the original comments to L80-81 (#referee #1) and L78-79 (referee #2): both referees point out that work by other modellers are incompletely or wrongly explained/cited. As a consequence you remove these references completely. However, I think this is not an adequate solution of this issue. Given that your manuscript should be published in GMD (i.e., in particular addressing model developers and users), I think it is quite essential that approaches of other models must be described and cited appropriately. Therefore, I kindly ask you to re-introduce a short paragraph with an improved (still short!) review of alternative model studies and model developments (i.e. based on others than your own model).

Response: We now have improved the paragraph (Line 74-97) with a review of other model studies and included the references (e.g., Spiegl et al., 2022; Sukhodolov et al., 2017) suggested by the reviewers.

**"The ability of general circulation models (GCMs, e.g., GISS ModelE, ECHAM5-HAM and EMAC) and chemical transport models (CTMs, e.g., GEOS-Chem and GMI ) to capture the main characteristics in $^7$Be and $^{10}$Be transport and deposition has been demonstrated in previous studies (e.g., Brattich et al., 2021; Field et al., 2006; Heikkilä et al., 2008; Koch & Rind, 1998; Liu et al., 2016; Spiegl et al., 2022; Sukhodolov et al., 2017; Usoskin et al., 2009). For example, Usoskin et al. (2009) found that the influence of the solar proton-induced $^7$Be production peak at the surface in early 2005 is small through the comparison of GISS ModelE simulations and surface air measurements. Heikkilä et al. (2009) showed that stratospheric $^{10}$Be contribution is dominant in the global $^{10}$Be deposition by tracing tropospheric and stratospheric $^{10}$Be separately in the aerosol-climate model ECHAM5-HAM. Spiegl et al. (2022) used the EMAC climate model to investigate the transport and deposition process of $^{10}$Be produced by the extreme solar proton event in 774/5 A.D. They suggested that the downward transport of $^{10}$Be from the stratosphere is mainly controlled by the Brewer-Dobson circulation in the stratosphere and cross-tropopause transport. By comparing the measurements with GEOS-Chem simulations over January-March 2003, Brattich et al. (2021) found that increased $^7$Be values in surface air samples in Northern Europe in early 2003 were associated with the instability of the Arctic polar vortex. They also showed that, while the model generally simulates well the month-to-month variation in surface $^7$Be concentrations, it tends to underestimate the observations (see their Table 2) partly due to the use of the default LP67 production rate for a solar maximum year (1958) in the GEOS-Chem model (Liu et al., 2001). By using the GMI CTM driven with four different meteorological datasets, Liu et al. (2016) showed that the observational constraints for $^7$Be and observed $^7$Be total deposition fluxes can be used to provide a first-order assessment of cross-tropopause transport in global models. In comparison to GCMs with or without nudged winds (e.g., Golubenko et al., 2021; Heikkilä et al., 2008; Spiegl et al., 2022) which involve simulating the entire global circulation and climate, the "offline" CTMs are driven by archived meteorological data sets, either from output of GCMs or from atmospheric data assimilation systems. For example, GEOS-Chem can be driven by the GEOS assimilated**

**meteorology (e.g., MERRA-2 reanalysis data; Gelaro et al., 2017) or output from the GISS GCM (e.g., Murray et al., 2021)"**

2. The second refers to the original comment to L327 (#referee #1): I think, this is an important information and should be added to the text. In particular you need to explain how the simulated stratospheric fraction of the 10Be deposition flux can be different from the simulated fraction of the 10Be concentration in the lowest model layer.

Response: Thank you for the comment. This was explained in the discussion of Fig. S4 (see line 364-367). We have added the reference Liu et al. (2016) to the sentence. Heikkilä et al. (2009) did not compare stratospheric fractions in total deposition fluxes vs surface concentrations and is thus not cited here. It has been cited elsewhere (see our response above).

**"Since the total deposition flux reflects precipitation scavenging through the tropospheric column, it tends to be more sensitive to $^7$Be air concentrations at higher altitudes and downward transport of $^7$Be from the stratosphere. Indeed, model results suggest that deposition fluxes have a higher stratospheric fraction compared to surface air concentrations (Fig. S4), as previously shown by Liu et al. (2016). "**

Brattich, E., Liu, H., Zhang, B., Hernández-Ceballos, M. Á., Paatero, J., Sarvan, D., et al. (2021). Observation and modeling of high-7Be events in Northern Europe associated with the instability of the Arctic polar vortex in early 2003. *Atmos. Chem. Phys. Discuss., 2021*, 1-43. https://doi.org/10.5194/acp-2020-1121

Field, C. V., Schmidt, G. A., Koch, D., & Salyk, C. (2006). Modeling production and climate-related impacts on $^{10}$Be concentration in ice cores. *Journal of Geophysical Research, 111*(D15). https://doi.org/10.1029/2005jd006410

Gelaro, R., McCarty, W., Suarez, M. J., Todling, R., Molod, A., Takacs, L., et al. (2017). The Modern-Era Retrospective Analysis for Research and Applications, Version 2 (MERRA-2). *J Clim, Volume 30*(Iss 13), 5419-5454. 10.1175/JCLI-D-16-0758.1

Golubenko, K., Rozanov, E., Kovaltsov, G., Leppänen, A.-P., Sukhodolov, T., & Usoskin, I. (2021). Application of CCM SOCOL-AERv2-BE to cosmogenic beryllium isotopes: description and validation for polar regions. *Geoscientific Model Development, 14*(12), 7605-7620. https://doi.org/10.5194/gmd-14-7605-2021

Heikkilä, U., Beer, J., & Feichter, J. (2008). Modeling cosmogenic radionuclides $^{10}$Be and $^7$Be during the Maunder Minimum using the ECHAM5-HAM General Circulation Model. *Atmospheric Chemistry and Physics, 8*(10), 2797-2809. https://doi.org/10.5194/acp-8-2797-2008

Heikkilä, U., Beer, J., & Feichter, J. (2009). Meridional transport and deposition of atmospheric $^{10}$Be. *Atmospheric Chemistry and Physics, 9*(2), 515-527. https://doi.org/10.5194/acp-9-515-2009

Koch, D., & Rind, D. (1998). Beryllium 10/beryllium 7 as a tracer of stratospheric transport. *Journal of Geophysical Research: Atmospheres, 103*(D4), 3907-3917. https://doi.org/10.1029/97JD03117

Liu, H., Considine, D. B., Horowitz, L. W., Crawford, J. H., Rodriguez, J. M., Strahan, S. E., et al. (2016). Using beryllium-7 to assess cross-tropopause transport in global models. *Atmospheric Chemistry and Physics, 16*(7), 4641-4659. https://doi.org/10.5194/acp-16-4641-2016

Liu, H., Jacob, D. J., Bey, I., & Yantosca, R. M. (2001). Constraints from $^{210}$Pb and $^7$Be on wet deposition and transport in a global three-dimensional chemical tracer model driven by assimilated meteorological fields. *Journal of Geophysical Research: Atmospheres, 106*(D11), 12109-12128. https://doi.org/10.1029/2000jd900839

Murray, L. T., Leibensperger, E. M., Orbe, C., Mickley, L. J., & Sulprizio, M. (2021). GCAP 2.0: a global 3-D chemical-transport model framework for past, present, and future climate scenarios. *Geosci. Model Dev., 14*(9), 5789-5823. 10.5194/gmd-14-5789-2021

Spiegl, T. C., Yoden, S., Langematz, U., Sato, T., Chhin, R., Noda, S., et al. (2022). Modeling the Transport and Deposition of $^{10}$Be Produced by the Strongest Solar Proton Event During the Holocene. *Journal of Geophysical Research: Atmospheres, 127*(13), e2021JD035658. https://doi.org/10.1029/2021JD035658

Sukhodolov, T., Usoskin, I., Rozanov, E., Asvestari, E., Ball, W. T., Curran, M. A., et al. (2017). Atmospheric impacts of the strongest known solar particle storm of 775 AD. *Sci Rep, 7*, 45257. https://doi.org/10.1038/srep45257

Usoskin, I. G., Field, C. V., Schmidt, G. A., Leppänen, A.-P., Aldahan, A., Kovaltsov, G. A., et al. (2009). Short-term production and synoptic influences on atmospheric $^{7}$Be concentrations. *Journal of Geophysical Research, 114*(D6), D06108. https://doi.org/10.1029/2008jd011333